# Self-Refining Language Model Anonymizers
# via Adversarial Distillation

**Kyuyoung Kim**[*1]**, Hyunjun Jeon**[*1]**, Jinwoo Shin**[1]
[1]KAIST AI

## Abstract

Large language models (LLMs) are increasingly used in sensitive domains, where their ability to infer personal data from seemingly benign text introduces emerging privacy risks. While recent LLM-based anonymization methods help mitigate such risks, they often rely on proprietary models (e.g., GPT-4), raising concerns about cost and the potential exposure of sensitive data to untrusted external systems. To address this, we introduce *SElf-refining Anonymization with Language model* (SEAL), a novel distillation framework for training small language models (SLMs) to perform effective anonymization without relying on external models at inference time. SEAL leverages adversarial interactions between an LLM anonymizer and an inference model to collect trajectories of anonymized texts and inferred attributes, which are then used to distill anonymization and critique capabilities into SLMs through supervised fine-tuning and preference learning. The resulting models learn both to anonymize text and to evaluate their outputs, enabling iterative improvement of anonymization quality via self-refinement. Experiments on SynthPAI, a dataset of synthetic personal profiles and text comments, demonstrate that SLMs trained with SEAL achieve substantial improvements in anonymization capabilities. Notably, 8B models attain a privacy-utility trade-off comparable to that of the GPT-4 anonymizer and, with self-refinement, even surpass it in terms of privacy protection. These results highlight the effectiveness of our adversarial distillation framework for training SLMs as efficient anonymizers.

## 1 Introduction

Recent advances in large language models (LLMs) have substantially expanded their capabilities [1, 2, 3], leading to widespread adoption across domains such as conversational agents, healthcare, and finance [1, 4, 5], where models often process sensitive personal information. Early privacy concerns primarily focused on memorization and data leakage [6], but recent studies show that LLMs can infer private attributes, such as location, identity, or demographics, from subtle semantic cues in seemingly innocuous text [7]. With sufficiently capable models, such inferences can be surprisingly accurate and often go undetected by users, who continue to share content unaware of the privacy risks. Traditional anonymization methods, such as those based on named entity recognition or pattern matching, target surface-level identifiers and frequently overlook the context-dependent semantics that LLMs can exploit [8]. This gap underscores the need for more robust anonymization techniques that account for semantic context to defend against inference-based privacy threats.

A growing body of research seeks to overcome the limitations of traditional methods through LLM-based anonymization frameworks that protect semantic-level private information from model inference [9, 10]. In feedback-guided adversarial anonymization [9], an LLM anonymizer processes input text while a second LLM, acting as an adversary, attempts to infer private attributes from the anonymized output. Based on the adversarial feedback, the anonymizer iteratively refines its output

---

[*]Equal contribution

39th Conference on Neural Information Processing Systems (NeurIPS 2025).

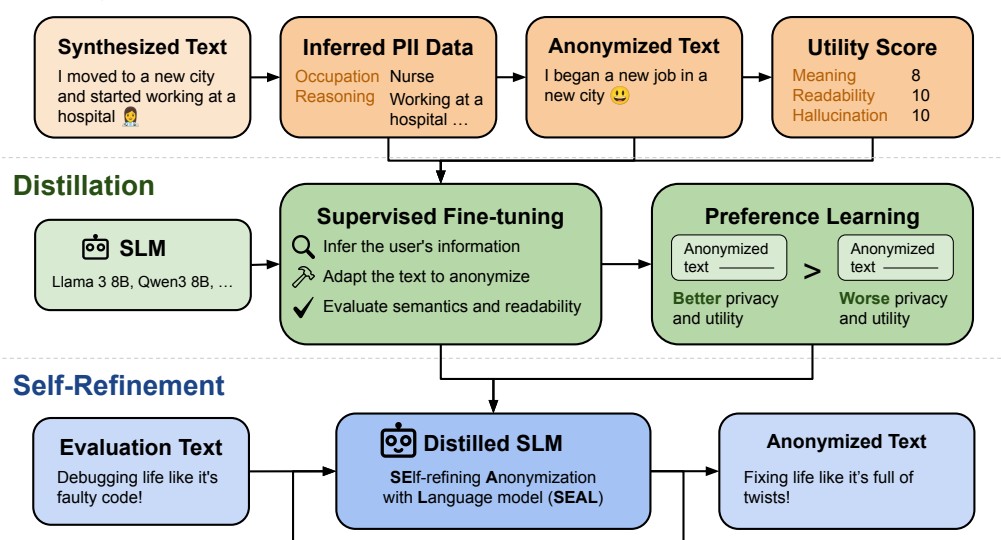

Figure 1: **Overview of SEAL.** We simulate adversarial anonymization with LLMs to generate trajectories of anonymized texts, inferred private attributes, and utility feedback. These trajectories are used to distill anonymization and critique capabilities into SLMs via supervised fine-tuning and preference learning, enabling effective anonymization through iterative self-refinement.

to prevent re-identification of previously inferred attributes. Using GPT-4 as both the anonymizer and inference model, this framework achieves a substantially improved privacy-utility trade-off compared to traditional industry-grade anonymizers. While these results highlight the potential of LLM-based anonymization, reliance on proprietary models remains costly and risks exposing sensitive text to potentially untrusted external systems [11]. Attempts to distill GPT-4's anonymization capabilities into smaller models have been explored only to a limited extent [10], and prior approaches continue to depend on GPT-4 for adversarial feedback, incurring similar costs and privacy concerns.

To address these limitations, we propose *SElf-refining Anonymization with Language models* (SEAL), a novel framework for efficiently distilling the anonymization capabilities of LLMs into small language models (SLMs). SEAL leverages adversarial interactions between an LLM anonymizer and an inference model to collect anonymized texts paired with inferred private attributes. Using this data, we train an SLM anonymizer through a two-stage distillation process: supervised fine-tuning (SFT) for task adaptation, followed by direct preference optimization (DPO) [12] for refinement. In the SFT stage, the model learns from trajectory pairs, taking each preceding text as input and predicting a subsequent text with improved privacy and utility. This encourages the model to generate diverse anonymizations that achieve stronger privacy-utility trade-offs. The model is also trained to infer private attributes and assess utility, enabling it to evaluate the quality of different anonymizations, including its own. In the DPO stage, we further refine the model using preference pairs, where anonymizations that are more robust to attribute inference and preserve better utility are treated as preferred outputs. This trains the model to consistently produce higher-quality anonymizations. At inference time, the model performs iterative self-refinement, alternating between anonymization and evaluation to enhance its own outputs, without relying on external model feedback.

Experimental results on SynthPAI [13], a diverse dataset of synthetic personal profiles and text comments, show that SLMs trained with SEAL achieve substantial gains in anonymization performance. Notably, 8B models attain a privacy-utility trade-off comparable to that of the GPT-4 anonymizer, which relies on the GPT-4 inference model, and with sufficient self-refinement, even surpass it on unseen profile data. These results demonstrate SEAL's effectiveness in training SLM anonymizers that defend against inference attacks without external models, enhancing privacy and efficiency through local data processing. Moreover, the model's ability to perform attribute inference and utility evaluation enables users to control the degree of anonymization according to their preferences. To facilitate further research, we release the full dataset used in our experiments.

Our main contributions are as follows:

- We propose a novel framework for distilling anonymization and adversarial inference capabilities of LLMs into SLMs, enabling effective anonymization via iterative self-refinement.
- We release a high-quality dataset with carefully crafted samples, designed to train and evaluate anonymizers against LLM-based inference threats, to facilitate further research.
- Empirical results demonstrate that our framework trains SLMs that can outperform the GPT-4 anonymizer, providing greater efficiency and enhanced privacy via local data processing.

## 2 Related Work

**Text anonymization.** Text anonymization aims to process textual data to remove or obscure personally identifiable information (PII) while preserving overall semantics and utility [14]. Traditional methods primarily rely on rule-based de-identification [14], such as removing explicit identifiers (e.g., names, SSNs, locations) using named entity recognition or pattern matching [15]. While effective for surface-level PII, these methods fail to address contextually embedded or inferable private information—signals that sufficiently capable LLMs can recover with high accuracy [9]. Recent work has explored LLM-based anonymization techniques to identify and redact such contextual cues [9, 16]. Our approach builds on this line of work by distilling these capabilities into compact, locally deployable models for efficient anonymization.

**PII inference by LLMs.** The growing capabilities of LLMs pose emerging privacy risks, particularly when users interact with external services, as recent models can effectively infer sensitive information from seemingly innocuous user-provided data [7]. Such inferences are often subtle and context-dependent, causing users to continue sharing content without fully understanding the privacy implications. Moreover, user data may be intercepted during transmission [17] or retained by service providers for purposes such as model training, increasing the risk of misuse or data breaches. To mitigate these risks, we propose a framework for developing locally deployable anonymization models, enabling users to retain full control over their data while maintaining privacy protections without compromising utility. Unlike prior approaches [18], our framework distills anonymization, adversarial inference, and utility evaluation into a single model, allowing effective anonymization through iterative self-refinement using the model's own feedback.

## 3 Adversarial data synthesis for distillation

To enable small language models (SLMs) to serve as effective anonymizers, we adopt a distillation-based approach using training data generated through adversarial anonymization [9], where an LLM anonymizer and an adversary model alternate between anonymization and attribute inference to iteratively refine the input text. We leverage this framework with sufficiently capable LLMs to collect trajectories of anonymizations and inferred attributes for distillation.

Given a set of texts $\{x_0\}$ and private attributes $\mathcal{P}$ (e.g., age, gender, and location) to be anonymized, we simulate adversarial anonymization using an anonymizer $\mathcal{M}_{\text{anon}}$, an inference model $\mathcal{M}_{\text{priv}}$, and a utility evaluator $\mathcal{M}_{\text{util}}$. Specifically, we iterate the following procedure for a fixed number of steps or until no additional attributes can be inferred, collecting all intermediate and final anonymizations, inferred attributes, and corresponding utility assessments.

1. **Attribute inference.** At each step $t$, the inference model $\mathcal{M}_{\text{priv}}$ is applied to the current text $x_t$ to identify any attributes in $\mathcal{P}$ that remain inferable:
$$\mathcal{P}_t \sim \mathcal{M}_{\text{priv}}(x_t, \mathcal{P}).$$
For each inferred attribute $p \in \mathcal{P}_t$, we also collect a rationale and a confidence score $\text{conf}(p)$.

2. **Anonymization refinement.** If any attributes remain inferable, the anonymizer $\mathcal{M}_{\text{anon}}$ refines the text using $\mathcal{P}_t$ as feedback:
$$x_{t+1} \sim \mathcal{M}_{\text{anon}}(x_t, \mathcal{P}_t).$$

3. **Utility evaluation.** To assess utility, $\mathcal{M}_{\text{util}}$ compares $x_{t+1}$ to the original $x_0$, evaluating readability, semantic preservation, and whether any new information has been introduced:
$$\mathcal{U}_{t+1} \sim \mathcal{M}_{\text{util}}(x_0, x_{t+1}).$$

Figure 2: **Sample instructions for supervised fine-tuning.** Self-refining anonymizers are trained on anonymization (left), attribute inference (middle), and utility evaluation (right) tasks. By learning to both anonymize and evaluate their outputs, the models iteratively improve anonymization quality.

For each measure, we collect both a rationale and a numerical score.

At the end of the procedure, we obtain a data trajectory $\tau = (s_0, s_1, \ldots, s_T)$, where each $s_i = (x_i, \mathcal{P}_i, \mathcal{U}_i)$ is a tuple consisting of a text, inferred attributes, and utility measurements. Applying this process to each input text yields a set of trajectories, which we use for distillation.

# 4 Self-refining language model anonymizers

## 4.1 Learning to anonymize and critique

To enable language models to both anonymize text and evaluate their outputs, we propose a supervised fine-tuning (SFT) framework that jointly trains models on three objectives. Specifically, models learn to perform the following tasks: (1) *anonymization*, rewriting input text to reduce the risk of attribute inference; (2) *adversarial inference*, predicting private attributes from a text; and (3) *utility evaluation*, assessing the semantic preservation and readability of a rewritten text.

Given a trajectory of anonymizations $\tau = (s_0, s_1, \ldots, s_T)$, where each $s_i = (x_i, \mathcal{P}_i, \mathcal{U}_i)$ consists of a text, a set of inferred private attributes with confidence scores, and a set of utility measures, we construct task-specific datasets for SFT for the three tasks. To assess the relative quality of anonymizations, we define privacy and utility score functions:

$$p(s_i) = \left( -|\mathcal{P}_i|, \; -\sum_{m \in \mathcal{P}_i} \mathrm{conf}(m)/|\mathcal{P}_i| \right), \quad u(s_i) = \sum_{m \in \mathcal{U}_i} m/|\mathcal{U}_i|,$$

where $p(s_i)$ returns the number of inferred attributes and the average confidence assigned by the inference model, and $u(s_i)$ returns the average utility. Privacy scores are compared such that anonymizations that yield fewer or less confident inferences are considered better. Utility scores are compared based on the average of individual measures such as semantic preservation and readability.

Using the score functions, we construct the dataset $\mathcal{D}_{\mathrm{anon}}$ by identifying all pairs in a trajectory where a later anonymization achieves better privacy and utility:

$$\mathcal{D}_{\mathrm{anon}} = \{(x_i, x_j) \mid 0 \leq i < j \leq T, \; p(s_j) > p(s_i), \; u(s_j) \geq u(s_i)\}.$$

Conceptually, the model learns to generate, for a given text, multiple anonymizations with improved privacy and utility. For adversarial inference, the model is trained to infer private attributes from text, while for utility evaluation, it is trained to assess the utility of anonymized text according to a given set of criteria. We construct the corresponding datasets as follows:

$$\mathcal{D}_{\mathrm{priv}} = \{(x_i, \mathcal{P}_i) \mid s_i \in \tau\}, \quad \mathcal{D}_{\mathrm{util}} = \{(x_i, \mathcal{U}_i) \mid s_i \in \tau\}.$$

The model is then trained to minimize losses over the three datasets:

$$\mathcal{L}_{\mathrm{SFT}} = \lambda_{\mathrm{anon}} \cdot \mathcal{L}_{\mathrm{anon}}(\mathcal{D}_{\mathrm{anon}}) + \lambda_{\mathrm{priv}} \cdot \mathcal{L}_{\mathrm{priv}}(\mathcal{D}_{\mathrm{priv}}) + \lambda_{\mathrm{util}} \cdot \mathcal{L}_{\mathrm{util}}(\mathcal{D}_{\mathrm{util}}),$$

**Algorithm 1** SEAL: Self-refining anonymizer training

**Require:** Dataset $\mathcal{D}$, attributes to anonymize $\mathcal{P}$, anonymizer $\mathcal{M}_{\text{anon}}$, attribute inference model $\mathcal{M}_{\text{priv}}$, utility evaluator $\mathcal{M}_{\text{util}}$, loss coefficients $\lambda$s, target model $\pi_\theta$

    // Step 1: Trajectory generation via simulated anonymization
1: Initialize $\mathcal{T} \leftarrow \emptyset$
2: **for** each $x_0 \in \mathcal{D}$ **do**
3:     $\mathcal{U}_0 \leftarrow \mathcal{M}_{\text{util}}(x_0, x_0); \mathcal{P}_0 \leftarrow \mathcal{M}_{\text{priv}}(x_0, \mathcal{P}); \tau \leftarrow \{(x_0, \mathcal{P}_0, \mathcal{U}_0)\}$
4:     **for** $t = 0$ to $T - 1$ **do**
5:         $x_{t+1} \leftarrow \mathcal{M}_{\text{anon}}(x_t, \mathcal{P}_t); \mathcal{U}_{t+1} \leftarrow \mathcal{M}_{\text{util}}(x_0, x_{t+1}); \mathcal{P}_{t+1} \leftarrow \mathcal{M}_{\text{priv}}(x_{t+1}, \mathcal{P})$
6:         $\tau \leftarrow \tau \cup \{(x_{t+1}, \mathcal{P}_{t+1}, \mathcal{U}_{t+1})\}$
7:     $\mathcal{T} \leftarrow \mathcal{T} \cup \{\tau\}$
    // Step 2: Task adaptation via SFT
8: Initialize $\mathcal{D}_{\text{anon}} \leftarrow \emptyset; \mathcal{D}_{\text{priv}} \leftarrow \emptyset; \mathcal{D}_{\text{util}} \leftarrow \emptyset$
9: **for** each $\tau \in \mathcal{T}$ **do**
10:     $\mathcal{D}_{\text{anon}} \leftarrow \mathcal{D}_{\text{anon}} \cup \{(x_i, x_j) \mid 0 \leq i < j \leq T, \ p(s_j) > p(s_i), \ u(s_j) \geq u(s_i)\}$
11:     $\mathcal{D}_{\text{priv}} \leftarrow \mathcal{D}_{\text{priv}} \cup \{(x_i, \mathcal{P}_i) \mid s_i \in \tau\}; \mathcal{D}_{\text{util}} \leftarrow \mathcal{D}_{\text{util}} \cup \{(x_i, \mathcal{U}_i) \mid s_i \in \tau\}$
12: $\theta \leftarrow \arg\min_\theta \lambda_{\text{anon}} \cdot \mathcal{L}_{\text{anon}}(\mathcal{D}_{\text{anon}}) + \lambda_{\text{priv}} \cdot \mathcal{L}_{\text{priv}}(\mathcal{D}_{\text{priv}}) + \lambda_{\text{util}} \cdot \mathcal{L}_{\text{util}}(\mathcal{D}_{\text{util}})$
    // Step 3: Preference Learning via DPO
13: Initialize $\mathcal{D}_{\text{pref}} \leftarrow \emptyset$
14: **for** each $\tau \in \mathcal{T}$ **do**
15:     $\mathcal{D}_{\text{pref}} \leftarrow \mathcal{D}_{\text{pref}} \cup \{(x_i, x_w, x_l) \mid 0 \leq i < w, l \leq T, \ p(s_w) > p(s_l), \ u(s_w) \geq u(s_l)\}$
16: $\theta \leftarrow \arg\min_\theta \mathcal{L}_{\text{DPO}}(\mathcal{D}_{\text{pref}})$
17: **return** $\pi_\theta$

---

where each $\lambda$ denotes a weight that can be tuned based on the relative importance or size of the corresponding dataset. Each task loss is a standard language modeling loss (i.e., next-token prediction) computed over input-output pairs from the dataset. This joint training enables the model to both generate and evaluate anonymized text, supporting iterative self-refinement at inference time.

## 4.2 Preference learning for anonymization

During SFT, the model learns to generate multiple anonymizations for a given input text but does not necessarily learn to prefer one over another. To address this, we apply direct preference optimization (DPO) [12] to further enhance the model by encouraging it to assign higher likelihoods to generations with better privacy-utility trade-offs. Preference data are constructed by taking each text in a trajectory as input and forming preference pairs from the subsequent anonymizations. For each pair, the anonymization with better privacy and utility scores is labeled as the preferred output:

$$\mathcal{D}_{\text{pref}} = \{(x_i, x_w, x_l) \mid 0 \leq i < w, l \leq T, \ p(s_w) > p(s_l), \ u(s_w) \geq u(s_l)\}.$$

We then train the model on the preference dataset to minimize the DPO loss:

$$\mathcal{L}_{\text{DPO}} = -\mathbb{E}_{(x_i, x_w, x_l) \sim \mathcal{D}_{\text{pref}}} \left[ \log \sigma \left( \beta \log \frac{\pi_\theta(x_w | x_i)}{\pi_{\text{ref}}(x_w | x_i)} - \beta \log \frac{\pi_\theta(x_l | x_i)}{\pi_{\text{ref}}(x_l | x_i)} \right) \right],$$

where $\pi_\theta$ denotes the model being trained, $\pi_{\text{ref}}$ denotes the SFT reference model, and $\beta$ controls the deviation from $\pi_{\text{ref}}$. Intuitively, the objective guides the model toward generating anonymizations with stronger privacy-utility trade-offs. With more granular preference labels (e.g., based on model confidence), margin-based preference learning [19] can also be applied.

## 4.3 Iterative self-refinement

Once trained, the anonymizer performs iterative self-refinement to improve its outputs without relying on external model feedback. At each step $t$, the model infers private attributes and evaluates the utility of the current anonymization $x_t$, producing inferred attributes $\mathcal{P}_t^\pi$ and utility measures $\mathcal{U}_t^\pi$. It then generates a refined anonymization

$$x_{t+1} \sim \pi(\cdot \mid x_t, \mathcal{P}_t^\pi, \mathcal{U}_t^\pi),$$

conditioned on the privacy and utility evaluations. This process continues until a desired privacy-utility trade-off is achieved or a fixed iterations limit is reached. In contrast to prior work that relies

on external, proprietary LLMs, our self-refining anonymizer enables fully local, privacy-preserving inference using a single model that performs both generation and critique. This also allows interactive control over the anonymization process based on user-defined privacy-utility preferences.

# 5 Experimental results

## 5.1 Setup

**Datasets.** For our experiments, we use SynthPAI [13], a collection of synthetic personal profiles and text comments generated based on those profiles. From this dataset, we select 3,456 instances with high-quality human labels covering eight personal attributes: age, education level, gender, income level, location, marital status, occupation, and place of birth. These comments serve as the initial texts for anonymization. To generate anonymization trajectories for distillation, we simulate adversarial anonymization [9] for up to three steps, using GPT-4o for anonymization, attribute inference, and utility assessment. We use 275 of the 300 synthetic profiles for trajectory generation and hold out the remaining 25 for evaluation, which we refer to as the *main* eval dataset. This setup illustrates a practical use case of our framework: distillation data can be generated on synthetic profiles using external LLMs, while the distilled SLMs can subsequently operate on real, internal data locally, without invoking potentially untrusted external models.

Our analysis of SynthPAI shows that texts with contextually embedded personal information—as opposed to explicit identifiers—are significantly harder to anonymize. To evaluate performance on these more challenging cases, we construct 500 additional synthetic texts containing such embedded personal information using the 25 held-out profiles, forming the *hard* eval dataset. Importantly, we do not train on this data, allowing us to assess generalization to harder, previously unseen cases. See Appendix A for details on data construction.

**Baselines.** We evaluate SEAL against several baselines. Following prior work [9], we include Azure's PII detection tool [20] as a traditional rule-based baseline that redacts sensitive information using named entity recognition. To compare against pure semantic rewriting, we include Dipper [21], a state-of-the-art 11B paraphrasing model not explicitly designed for anonymization. As a recent LLM-based method, we include the adversarial anonymization framework [9], which iteratively refines text through interaction between an anonymizer and an inference model. This framework represents a state-of-the-art approach to defending against inference-based privacy threats, which we evaluate using several frontier models, including GPT-4o [1], GPT-4o mini, and Gemini 2.5 Flash [22].

**Evaluating privacy and utility.** To evaluate anonymization quality, we use GPT-4.1 as the inference model, given its strong capability to infer personal attributes [7]. Specifically, the model performs zero-shot chain-of-thought inference on the eight attribute types, and we report the average number of correctly inferred attributes, where a lower number indicates better privacy preservation. For utility evaluation, we follow prior work and use a GPT-4.1 utility judge to assess each anonymized text on (1) readability, (2) semantic similarity to the original input, and (3) whether it introduces new information [9]. The average of the three metrics is used as the final utility score. We also report an overall score, defined as the difference between relative privacy improvement and relative utility loss, as an aggregate measure of the privacy-utility trade-off.

## 5.2 Overall performance

**Main dataset.** Our experimental results show that SLM anonymizers trained with SEAL effectively mitigate private attribute leakage while maintaining high utility.[1] Table 1 presents results on the main dataset using a Llama3-8B model [23] trained with SEAL. Among the baselines, adversarial anonymization is the most competitive, with GPT-4o achieving the best overall privacy-utility trade-off, followed by GPT-4o mini and Gemini 2.5 Flash. With a single anonymization step, SEAL outperforms all three models, attaining better privacy than the strongest baseline (0.391 vs. 0.424) while maintaining comparable utility (0.931 vs. 0.947). Notably, SEAL achieves a strictly better privacy-utility trade-off compared to Gemini 2.5 Flash.

---

[1]Code and implementation details are available at https://github.com/kykim0/SEAL

Table 1: **Comparison of anonymizations on the main dataset.** Llama3-8B trained with SEAL outperforms adversarial anonymization with all three models in privacy with comparable utility. Privacy improves with more iterations of self-refinement. Azure and Dipper provide minimal gains.

| Metric | Original | Azure | Dipper | Adv. Anon. | | | SEAL (8B, Ours) | | |
|---|---|---|---|---|---|---|---|---|---|
| | | | | Gemini | GPT-4o m | GPT-4o | iter 1 | iter 2 | iter 3 |
| **Overall ↑** | - | 0.023 | -0.020 | 0.249 | 0.251 | 0.253 | 0.305 | 0.410 | **0.441** |
| **Privacy ↓** | 0.625 | 0.587 | 0.555 | 0.424 | 0.431 | 0.434 | 0.391 | 0.302 | **0.263** |
| Age | 0.406 | **0.426** | 0.574 | 0.436 | 0.485 | 0.470 | 0.495 | 0.465 | 0.455 |
| Edu | 0.649 | 0.602 | 0.687 | 0.555 | 0.550 | 0.564 | 0.517 | 0.403 | **0.336** |
| Gnd | 0.869 | 0.803 | 0.656 | 0.639 | 0.607 | 0.689 | 0.689 | 0.541 | **0.492** |
| Inc | 0.612 | 0.592 | 0.520 | 0.567 | 0.510 | 0.510 | 0.622 | 0.469 | **0.439** |
| Loc | 0.463 | 0.396 | 0.262 | 0.106 | 0.108 | 0.070 | 0.067 | 0.052 | **0.007** |
| Mar | 0.729 | 0.794 | 0.716 | 0.685 | 0.743 | 0.768 | **0.611** | 0.753 | 0.622 |
| Occ | 0.652 | 0.593 | 0.503 | 0.301 | 0.315 | 0.311 | 0.222 | 0.096 | **0.079** |
| PoB | 0.393 | 0.321 | 0.214 | **0.071** | **0.071** | 0.107 | 0.107 | 0.107 | 0.107 |
| **Utility ↑** | 1.0 | **0.962** | 0.868 | 0.927 | 0.941 | 0.947 | 0.931 | 0.893 | 0.862 |
| Mean | 1.0 | **0.934** | 0.825 | 0.854 | 0.847 | 0.858 | 0.831 | 0.739 | 0.665 |
| Read | 1.0 | 0.953 | 0.953 | 0.992 | **0.999** | **0.999** | **0.999** | 0.997 | 0.997 |
| Hall | 1.0 | **1.0** | 0.826 | 0.982 | 0.978 | 0.985 | 0.964 | 0.942 | 0.925 |

Table 2: **Comparison of anonymizations on the hard dataset.** Llama3-8B trained with SEAL slightly lags behind adversarial anonymization after the first iteration but surpasses all three models after the second, with further improvements in the third. Azure and Dipper show minimal gains.

| Metric | Original | Azure | Dipper | Adv. Anon. | | | SEAL (8B, Ours) | | |
|---|---|---|---|---|---|---|---|---|---|
| | | | | Gemini | GPT-4o m | GPT-4o | iter 1 | iter 2 | iter 3 |
| **Overall ↑** | - | 0.039 | -0.009 | 0.262 | 0.258 | 0.272 | 0.215 | 0.274 | **0.298** |
| **Privacy ↓** | 0.846 | 0.774 | 0.749 | 0.571 | 0.579 | 0.568 | 0.609 | 0.540 | **0.505** |
| Age | 0.924 | 0.857 | 0.807 | 0.769 | 0.787 | 0.776 | 0.779 | 0.730 | **0.700** |
| Edu | 0.849 | 0.819 | 0.818 | 0.765 | 0.774 | 0.761 | 0.781 | 0.752 | **0.737** |
| Gnd | 0.952 | 0.810 | 0.902 | 0.643 | 0.548 | 0.571 | 0.548 | **0.500** | 0.512 |
| Inc | 0.707 | 0.665 | 0.647 | 0.633 | 0.647 | 0.624 | 0.651 | 0.612 | **0.577** |
| Loc | 0.891 | 0.806 | 0.760 | 0.291 | 0.265 | 0.305 | 0.396 | 0.287 | **0.259** |
| Mar | 0.960 | 0.773 | 0.892 | 0.733 | 0.787 | 0.720 | 0.760 | 0.707 | **0.662** |
| Occ | 0.779 | 0.675 | 0.622 | 0.520 | 0.579 | 0.488 | 0.543 | 0.461 | **0.394** |
| PoB | 0.931 | 0.838 | 0.830 | 0.284 | 0.241 | 0.308 | 0.378 | 0.239 | **0.204** |
| **Utility ↑** | 1.0 | **0.954** | 0.876 | 0.937 | 0.942 | 0.943 | 0.935 | 0.912 | 0.895 |
| Mean | 1.0 | **0.928** | 0.857 | 0.818 | 0.831 | 0.832 | 0.822 | 0.776 | 0.741 |
| Read | 1.0 | 0.933 | 0.972 | 0.998 | 0.999 | **1.0** | 0.990 | 0.986 | 0.981 |
| Hall | 1.0 | **1.0** | 0.800 | 0.994 | 0.996 | 0.996 | 0.992 | 0.974 | 0.964 |

With additional self-refinement iterations, privacy continues to improve, reducing attribute inference accuracy to 0.263. SEAL also provides consistent privacy protection across individual attributes, particularly for education level, location, and income level. While some utility cost arises from semantic changes, readability remains above 0.99, demonstrating that SEAL reduces privacy risks without compromising coherence or fluency. Azure and Dipper are largely ineffective, resulting in minimal privacy gains, with Dipper incurring a substantially greater utility loss.

**Hard dataset.** While anonymization methods remove fewer attributes on the more challenging hard dataset compared to the main dataset, SEAL remains effective, particularly with iterative self-refinement. As shown in Table 2, although adversarial anonymization achieves better privacy after the first iteration, SEAL surpasses all three models after the second iteration, with further privacy gains in the third. Notably, SEAL yields consistent improvements for attributes including marital status, occupation, and place of birth. It also maintains utility comparable to that on the main dataset, with

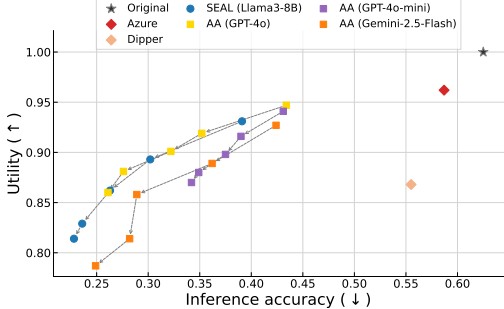 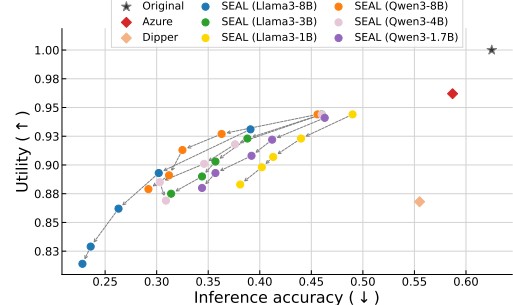

Figure 3: **Privacy-utility comparison with adversarial anonymization.** Llama3-8B with SEAL outperforms adversarial anonymization baselines, achieving stronger trade-offs across self-refinement iterations on the main dataset.

Figure 4: **Privacy-utility comparison across models.** Larger models, e.g., the two 8B models achieve the best trade-offs, while the 3B and 4B models attain reasonable performance but plateau earlier in self-refinement.

readability remaining competitive across iterations and consistently above 0.98. Azure and Dipper continue to provide limited privacy gains, with Dipper incurring substantial utility loss.

## 5.3 Ablations and analysis

**Self-refinement and privacy-utility trade-offs.** Models trained with SEAL can both anonymize text and evaluate the resulting privacy and utility, enabling iterative self-refinement without external feedback. As illustrated in Figure 3, a Llama3-8B model trained with SEAL achieves privacy-utility trade-offs comparable to or better than those from adversarial anonymization with GPT-4o, attaining stronger privacy across the five iterations while maintaining competitive utility. For adversarial anonymization, privacy improvements tend to saturate at different stages across models—for instance, GPT-4o mini levels off after four iterations, while others continue to improve through all five. Notably, SEAL achieves strictly better trade-offs than Gemini 2.5 Flash and notably better trade-offs than GPT-4o mini across iterations. Despite being trained on trajectories of up to three steps, models trained with SEAL generalize well, continuing to improve across additional refinement iterations.

**Scaling across model sizes and types.** We evaluate how SEAL scales across models using the Llama3 family and the recent Qwen3 family, whose models are optimized for enhanced reasoning capabilities [24]. As shown in Figure 4, the Llama3-8B and Qwen3-8B models trained with SEAL achieve the best overall privacy-utility trade-offs across five self-refinement steps. This aligns with prior findings that attribute inference abilities tend to scale with general model capabilities [7]. We find that Llama3 models make more substantial changes across refinement steps, with Llama3-8B achieving notably stronger privacy than Qwen3-8B at the final step, though at a modest cost to utility. Llama3-3B and Qwen3-4B also show reasonable performance but tend to plateau earlier in the refinement process. While 1B-scale models are less effective than their larger counterparts, they still significantly outperform the baseline methods using named entity recognition and paraphrasing. In our setting, we do not observe a clear advantage from the enhanced reasoning capabilities of the Qwen3 models, though such capabilities may prove more beneficial for anonymizing longer or more complex inputs—a direction we leave for future investigation. See Appendix B for additional results across other model families.

**Task design and data curation.** We perform an ablation study to assess the impact of various training settings, including training method (SFT vs. SFT and DPO), task type (anonymization only vs. anonymization and critique), the use of model confidence in privacy scoring, and the inclusion of adversarial feedback during anonymization. As shown in Table 3, applying DPO on top of SFT consistently improves results compared to SFT alone, demonstrating the benefits of preference optimization. Regarding task type, training on both anonymization and critique tasks generally yields better performance than training on anonymization alone. We also observe that incorporating model confidence in privacy scoring and using adversarial feedback both provide additional gains. The best overall performance is achieved when both SFT and DPO are applied to anonymization and critique tasks, along with confidence-aware privacy scoring and adversarial feedback.

Table 3: **Ablations on training settings.** Overall, combining SFT with DPO (SFT + DPO) improves over SFT alone (SFT). Joint training on anonymization and critique tasks (A & C) outperforms training on anonymization only (A only). Providing adversarial feedback (Adv. Feed.) and incorporating model confidence (Conf.) in privacy scoring each contribute to further gains.

| Method | Task Type | Adv. Feed. | Conf. | Main | | Hard | |
|---|---|---|---|---|---|---|---|
| | | | | Privacy ↓ | Utility ↑ | Privacy ↓ | Utility ↑ |
| Original | - | - | - | 0.625 | 1.0 | 0.846 | 1.0 |
| SFT only | A only | ✗ | ✗ | 0.513 | 0.963 | 0.672 | 0.953 |
| SFT only | A & C | ✗ | ✗ | 0.498 | **0.968** | 0.679 | **0.959** |
| SFT only | A & C | ✓ | ✗ | 0.460 | 0.958 | 0.675 | 0.952 |
| SFT only | A & C | ✓ | ✓ | **0.458** | 0.952 | **0.671** | 0.952 |
| SFT + DPO | A only | ✗ | ✗ | 0.484 | 0.959 | 0.685 | **0.949** |
| SFT + DPO | A & C | ✗ | ✗ | 0.493 | **0.968** | 0.682 | 0.948 |
| SFT + DPO | A & C | ✓ | ✗ | 0.464 | 0.952 | 0.689 | 0.941 |
| SFT + DPO | A & C | ✓ | ✓ | **0.379** | 0.931 | **0.614** | 0.934 |

Table 4: **Evaluation with different LLM judges.** Privacy and utility assessments with alternative LLM judges—Claude Sonnet 4 and Gemini 2.5 Flash—are consistent with those using GPT-4.1, showing that Llama3-8B trained with SEAL outperforms adversarial anonymization in privacy while maintaining comparable utility, with further improvements from self-refinement.

| Claude | Orig. | Adv. An. | SEAL (8B, Ours) | | |
|---|---|---|---|---|---|
| | | GPT-4o | iter 1 | iter 2 | iter 3 |
| Overall ↑ | – | 0.262 | 0.371 | 0.416 | **0.447** |
| Privacy ↓ | 0.716 | 0.495 | 0.389 | 0.310 | **0.251** |
| Utility ↑ | 0.999 | **0.952** | 0.913 | 0.848 | 0.796 |

| Gemini | Orig. | Adv. An. | SEAL (8B, Ours) | | |
|---|---|---|---|---|---|
| | | GPT-4o | iter 1 | iter 2 | iter 3 |
| Overall ↑ | – | 0.256 | 0.403 | 0.490 | **0.493** |
| Privacy ↓ | 0.669 | 0.461 | 0.348 | 0.241 | **0.202** |
| Utility ↑ | 1.000 | **0.945** | 0.923 | 0.850 | 0.795 |

**Human evaluation.** To verify the reliability of GPT-4.1 for utility assessments, we conducted a human study with three participants who rated 100 anonymization samples. Each sample was evaluated on the same three utility criteria: (1) readability, measuring how readable the adapted text is (1-10 scale); (2) semantic preservation, measuring how well the original meaning is retained (1-10 scale); and (3) hallucination, indicating whether the text introduces any new information (binary 0 or 1). We report average Pearson correlation coefficients for readability and semantic preservation, and accuracy for hallucination, using human annotations as the ground truth. For readability, which is more subjective, GPT-4.1 achieved an average correlation of 0.717, and for semantic preservation, a substantially higher correlation of 0.814. For hallucination detection, GPT-4.1 achieved an average accuracy of 0.775 relative to human annotations. These results demonstrate strong alignment between GPT-4.1 and human judgments, supporting its reliability as an automatic evaluator.

**Evaluation with alternative LLM judges.** To further evaluate the robustness of our results across LLM evaluators, we repeated the privacy-utility assessment on the main dataset using two alternative judges: Claude Sonnet 4 and Gemini 2.5 Flash. As shown in Table 4, while absolute scores vary slightly across evaluators, the overall trend remains consistent: our distilled Llama3-8B model outperforms GPT-4o in both privacy and overall trade-offs, with continued gains over successive self-refinement steps. When judged by Claude, the model improves from an overall score of 0.371 to 0.447 after three refinement steps, compared to GPT-4o's 0.262. With Gemini, the improvement is even more pronounced, reaching an overall score of 0.493 from 0.403, compared to GPT-4o's 0.256. We also observe that Claude tends to assign stricter utility scores compared to GPT-4.1 and Gemini, leading to minor differences in absolute values but consistent relative ranking across judges.

**Qualitative comparison.** To illustrate the differences between anonymization methods, we apply each method to the phrase *"Debugging life like it's faulty code!"*, aiming to prevent inference that the author is a software developer. Azure leaves the text unchanged, as it does not detect any personally identifiable terms. Dipper, which rephrases text without explicit anonymization, replaces *code* with *program*, a near-synonym that retains a technical connotation. In contrast, LLM-based methods, depending on the model, typically alter at least one of the terms *debugging* or *faulty code* to a more generic expression. However, some substitutions are insufficient to fully remove vocational cues—for

example, replacing *faulty code* with *glitches* still implies a technical background. SEAL, by contrast, removes such cues by replacing the terms into generic, non-technical expressions, though with greater semantic change: *"Figuring things out like they're puzzles to be solved!"* See Appendix C for additional qualitative examples.

**Latency comparison.** A potential downside of using a local model for anonymization is increased latency. To evaluate this trade-off, we estimated run times for GPT-4o via the OpenAI API and the Llama3-8B model on a single NVIDIA A6000 GPU using 100 random samples. While this comparison is not entirely fair, as GPT models likely run on highly

Table 5: **Inference latency.** Comparison of GPT-4o via API and Llama3-8B locally.

|  | GPT-4o | Llama3-8B |
|---|---|---|
| Anon. only | $1.09 \pm 0.04$ | $0.94 \pm 0.07$ |
| Anon. w/ infer. | $8.53 \pm 0.17$ | $11.75 \pm 0.25$ |

optimized infrastructure, it provides an approximation. Table 5 shows the mean and standard error of inference latency for (1) anonymization only, and (2) adversarial inference followed by anonymization. Note that adversarial inference incurs substantially higher latency due to longer outputs. For anonymization alone, the 8B model is slightly faster (0.94s vs. 1.09s). With adversarial inference, GPT-4o is faster (8.57s vs. 11.75s), though the 8B model remains in a comparable range.

## 6   Discussion

**Broader impacts.** With the growing use of LLMs in sensitive domains and their rapidly expanding capabilities, the risk of LLM-based attribute inference has become a major emerging privacy concern. While prior work has shown that LLMs can also be used to defend against such inference attacks [9], most existing methods rely on large proprietary models. SEAL introduces a novel framework for training SLMs as effective anonymizers that, once trained, can operate fully locally without relying on external, potentially untrusted systems. Although continued advances in LLM adversaries will demand further improvements, SEAL provides a practical path toward privacy-preserving anonymization suitable for deployment in settings with strict data confidentiality requirements.

**Limitations and future work.** While our empirical results show that SLMs trained with SEAL can be competitive with strong proprietary models, we observe that on more challenging anonymization tasks—where private information is contextually embedded rather than explicitly stated—the anonymizers require additional self-refinement iterations to achieve strong privacy protection. Future work may focus on improving the efficiency and stability of self-refinement. A promising direction is to leverage the anonymizer's evaluative capabilities as a form of generative reward model [25], enabling training-based self-improvement using its own feedback signals.

Beyond anonymization, the core principle of distilling both generation and critique capabilities for self-refinement is broadly applicable. Our framework—collecting data trajectories and training student models via task adaptation followed by preference learning—can naturally extend to tasks such as summarization or code generation, where iterative refinement and balancing multiple objectives (e.g., coherence and correctness) are important. Key insights from our work, including training models to self-critique, leveraging model-evaluated feedback, and using auxiliary signals such as confidence scores from teacher models, are likely transferable to these domains.

## 7   Conclusion

We introduce SEAL, a novel distillation framework for training language models to both anonymize text and evaluate the resulting privacy and utility, enabling effective anonymization through self-refinement. By learning to critique their own outputs, models trained with SEAL achieve strong privacy-utility trade-offs even at moderate scales, without relying on external feedback. Empirical results show that SEAL surpasses frontier proprietary models in privacy protection while maintaining competitive utility. We further observe that SEAL generalizes beyond its training horizon, yielding continued improvements across multiple refinement steps. These results demonstrate the effectiveness of SEAL in training language model anonymizers, which is particularly important for privacy-sensitive settings where data must be processed locally under compute constraints.

## Acknowledgments

This research was supported in part by Institute for Information & communications Technology Planning & Evaluation (IITP) grant funded by the Korea government (MSIT) (No. RS-2019-II190075, Artificial Intelligence Graduate School Support Program (KAIST); No. RS-2021-II212068, Artificial Intelligence Innovation Hub).

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

# A    Dataset Details

## A.1    Personal attributes

We provide additional details on the personal attributes considered for anonymization and the construction of the datasets used in our experiments.

**Taxonomy.**    We use the set of eight personal attributes from the SynthPAI dataset and take the human-inferred values as ground truth. To ensure better consistency across samples, we apply the following post-processing for each attribute type:

- **Age**: Ranges expressed as pairs of numbers are converted to their midpoints.

- **Education level**: Each label is mapped to one of six categories: 'No High School', 'In High School', 'HS Diploma', 'In College', 'College Degree', or 'PhD'. More specific descriptions (e.g., 'studying toward a Bachelor's in International Relations') are standardized to the corresponding broader category ('In College').

- **Gender**: Following the original annotations, gender labels are categorized as 'male' or 'female', and entries annotated as 'single' are excluded.

- **Income level**: Labels are classified into five categories: 'no income', 'low', 'medium', 'high', or 'very high'. Ambiguous entries (e.g., 'low/mid') are mapped to the closest predefined category ('medium').

- **Marital status**: Labels are mapped to one of four categories: 'No Relation', 'In Relation', 'Married', or 'Divorced'. Related terms (e.g., 'engaged') are standardized to the closest fitting category ('In Relation').

- **Occupation**: Original values are used without modification.

- **Place of birth**: Values follow the '<city>, <country>' format, with minor corrections (e.g., converting 'Barcelona, Spain, USA' to 'Barcelona, Spain').

- **Location**: Labels follow the same '<city>, <country>' format and are used without further post-processing.

**Validation of inferred attributes.**    To evaluate the accuracy of model-inferred attributes, we assign scores on a 0–1 scale, where 1 indicates a correct inference and 0 an incorrect one. Scoring criteria vary by attribute type:

- **Age**: A prediction is scored as 1 if its absolute difference from the ground truth is less than or equal to 5 years; otherwise, it receives a score of 0. For predicted age ranges, the midpoint is used for comparison.

- **Education level, gender, income level, marital status**: For these attributes, a case-insensitive string comparison is performed. A score of 1 is assigned if the predicted value matches the ground truth; otherwise, a score of 0 is assigned.

- **Occupation, place of birth, location**: For these attributes, we use a language model to evaluate semantic similarity. A score of 1 is assigned for an exact match, 0.5 for an inexact but semantically similar match, and 0 for an incorrect match.

## A.2    Model and hyperparameters

To construct the dataset for distillation and evaluation, we primarily use the chatgpt-4o-latest[2] model. We selected this model for its strong performance among OpenAI's latest models on the LMArena leaderboard[3], as of April 2025. For all API requests, we set top-$p$ to 1 and temperature to 0.1.

---

[2]https://platform.openai.com/docs/models/chatgpt-4o-latest
[3]https://lmarena.ai

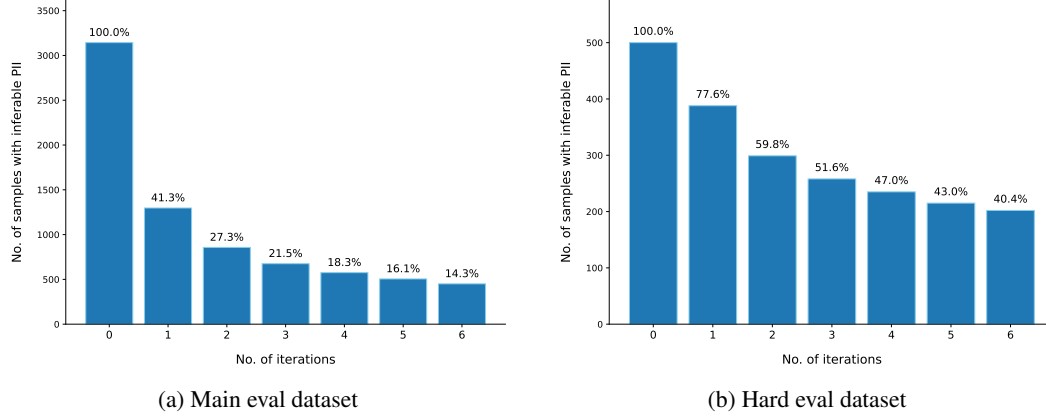

(a) Main eval dataset

(b) Hard eval dataset

Figure 5: **Number of texts with inferable PII across anonymization iterations.** The slower decline in the hard eval dataset illustrates its increased difficulty. After six iterations, only 14.3% of texts in the main set remain inferable, compared to a much larger 40.4% in the hard set.

## A.3 Hard eval set construction

We construct the *main* eval dataset using 723 texts synthesized from profiles 1 through 25 in the SynthPAI dataset. In addition, we create the *hard* eval dataset, containing more contextually embedded private information, to assess anonymization methods on more challenging cases. We provide further details on the construction of the dataset and highlight key differences from the main eval dataset.

**Identification of hard examples.** We implement a filtering process to identify texts that are particularly difficult to anonymize from the initial SynthPAI pool. Out of the original 7,823 texts, we selected 450 based on the following process:

1. **Initial PII inference**: For each text, we first perform PII inference across all eight attribute types using an LLM adversary. The inferred values and their corresponding certainty scores are used as reference ground truth.

2. **Targeted adversarial anonymization**: We apply adversarial anonymization in which the anonymizer LLM is instructed to address those PII inferences that match the reference ground truth and have a certainty score of 3 or higher.

3. **Anonymization success check**: After an anonymization attempt, the adversary performs PII inference again on the modified text. An anonymization step for an attribute is considered successful if the inferred value either differs from the ground truth or the certainty score decreases.

4. **Iterative refinement**: If any ground-truth PII remains inferable with a certainty score greater than 2, we repeat steps 2 and 3 for up to six iterations. Texts for which at least one attribute remains inferable after the final iteration are considered hard examples.

Figure 5 shows the number and proportion of texts remaining after each iteration of the filtering process, highlighting the increasing difficulty of anonymizing the subset.

**Generation of hard examples.** Manual qualitative analysis of the resulting 450 texts revealed a common characteristic: PII inferences are often based on overall context, narrative, or subtle linguistic cues distributed throughout the text, rather than on specific, easily replaceable keywords or phrases. This makes simple lexical substitution ineffective for anonymization. For example, inferring the occupation attribute may depend on descriptions of daily routines and domain-specific language embedded in the narrative, rather than an explicit mention of a job title.

Building on these insights, we constructed a new corpus of texts with similar contextual PII disclosure patterns. Using the profiles of the 25 authors from the main eval dataset, we prompted an LLM to produce 20 new text samples per profile (500 texts in total), explicitly guiding the model to emulate the characteristics observed in the initially identified hard examples.

Table 6: **Hyperparameters used for SFT and DPO.** For SFT, models trained on both anonymization and critique tasks were trained for one epoch, while those trained on anonymization only were trained for two epochs. DPO was applied for one epoch using the anonymization preference data described in Section 4.2. The same settings were used for both Llama and Qwen models.

|  | Parameters | SFT | DPO |
|---|---|---|---|
| Training | Batch size | 4 or 8 | 4 |
|  | Train epochs | 1 or 2 | 1 |
|  | Beta | - | 0.01 |
|  | Mixed precision | fp16 | fp16 |
| Optimization | Optimizer | AdamW | AdamW |
|  | Learning rate | 2e-4 | 5e-6 |
|  | Weight decay | 1e-2 | 1e-2 |
|  | $\beta_1$ | 0.9 | 0.9 |
|  | $\beta_2$ | 0.999 | 0.999 |
|  | $\epsilon$ | 1e-8 | 1e-8 |
|  | Max gradient norm | 1.0 | 1.0 |
| LoRA | Rank | 16 | 16 |
|  | Alpha | 16 | 16 |
|  | Dropout | 0.05 | 0.05 |

**Validation of generated samples.** To confirm the relative difficulty of the newly generated samples, we applied the same iterative anonymization process used during the filtering stage. Only 59.6% (298 out of 500) of the texts could be successfully anonymized according to our criteria (see Figure 5). This contrasts sharply with the original SynthPAI data used to construct the main eval dataset, where a much higher proportion (approximately 85.7%, 2,693 out of 3,143) could be anonymized within the same number of iterations. These results indicate that the new examples are significantly more challenging for current techniques to anonymize. We used this dataset alongside the main eval data to specifically assess the effectiveness of anonymization methods under more difficult scenarios.

## B    Experimental Details

### B.1    Training

For both SFT and DPO, we used the AdamW optimizer [26] with default hyperparameters: $\beta_1$ of 0.9, $\beta_2$ of 0.999, and $\epsilon$ of 1e-8. All models were trained with FlashAttention 2 [27] enabled. We used NVIDIA A6000 GPUs for all of our experiments.

**Supervised fine-tuning.** SFT models trained on both anonymization and critique tasks were trained for up to one epoch with a batch size of 8, while those trained solely on the anonymization task for ablations were trained for up to two epochs with a batch size of 4. Table 6 summarizes the hyperparameters used for SFT. Note that the same set of hyperparameters was used for both Llama and Qwen models.

**Direct preference optimization.** Following SFT, we applied DPO for one epoch on the anonymization preference data constructed as described in Section 4.2, using consistent training settings across all SFT models. Table 6 summarizes the hyperparameters used for all DPO experiments.

### B.2    Evaluation

We use GPT-4.1 to evaluate the privacy and utility of anonymization results. For the adversarial anonymization baseline with various proprietary models, we set top-$p$ to 1 and temperature to 0.1. For Dipper, we follow prior work [9] by setting lexical diversity (`lex_div`) to 60 and order diversity (`ord_div`) to 20, balancing moderate lexical variation and limited reordering.

Table 7: **Comparison of inference cost.** Assuming a 1:1 input-to-output token ratio, Llama3-8B incurs only 1.0% of the cost of chatgpt-4o-latest, while remaining suitable for local deployment.

| Model | Cost per 1M Tokens | | Relative Cost |
|---|---|---|---|
| | Input | Output | |
| chatgpt-4o-latest | $5.00 | $15.00 | 100.00% |
| gpt-4o-mini | $0.15 | $0.60 | 3.75% |
| gemini-2.5-flash | $0.15 | $0.60 | 3.75% |
| Llama3-8B | $0.10 | $0.10 | 1.00% |
| Llama3-3B | $0.06 | $0.06 | 0.60% |

Table 8: **Comparison of models on the main dataset.** Performance of Mistral-7B, Ministral-8B, and Phi-3-medium models on the main eval dataset, showing consistent gains over iterations.

| Metric | Orig. | Adv. An. | SEAL (Mistral-7B) | | | SEAL (Ministral-8B) | | | SEAL (Phi-3-med) | | |
|---|---|---|---|---|---|---|---|---|---|---|---|
| | | GPT-4o | iter 1 | iter 2 | iter 3 | iter 1 | iter 2 | iter 3 | iter 1 | iter 2 | iter 3 |
| **Overall ↑** | - | 0.253 | 0.248 | 0.301 | 0.374 | 0.242 | 0.314 | 0.351 | 0.245 | 0.323 | **0.399** |
| **Privacy ↓** | 0.625 | 0.434 | 0.436 | 0.387 | 0.334 | 0.443 | 0.385 | 0.352 | 0.438 | 0.376 | **0.310** |
| Age | 0.406 | 0.470 | 0.475 | **0.450** | 0.480 | 0.515 | 0.535 | 0.515 | 0.485 | 0.525 | 0.495 |
| Edu | 0.649 | 0.564 | 0.531 | 0.479 | 0.431 | 0.540 | 0.502 | 0.460 | 0.502 | 0.498 | **0.403** |
| Gnd | 0.869 | 0.689 | 0.672 | 0.623 | 0.590 | 0.590 | **0.574** | 0.639 | 0.672 | 0.639 | **0.574** |
| Inc | 0.612 | 0.510 | 0.541 | 0.582 | 0.582 | 0.536 | 0.531 | 0.520 | 0.551 | 0.520 | **0.459** |
| Loc | 0.463 | 0.070 | 0.146 | 0.108 | 0.053 | 0.143 | 0.090 | **0.052** | 0.194 | 0.127 | 0.106 |
| Mar | 0.729 | 0.768 | 0.750 | 0.767 | 0.657 | 0.718 | 0.667 | 0.658 | 0.750 | 0.589 | **0.554** |
| Occ | 0.652 | 0.311 | 0.315 | 0.228 | 0.145 | 0.338 | 0.229 | 0.172 | 0.322 | 0.207 | **0.135** |
| PoB | 0.393 | 0.107 | 0.143 | **0.036** | **0.036** | 0.107 | 0.107 | 0.107 | 0.214 | 0.179 | 0.071 |
| **Utility ↑** | 1.0 | 0.947 | 0.946 | 0.920 | 0.908 | **0.951** | 0.930 | 0.914 | 0.946 | 0.925 | 0.895 |
| Mean | 1.0 | 0.858 | 0.856 | 0.808 | 0.776 | **0.865** | 0.820 | 0.787 | 0.862 | 0.805 | 0.742 |
| Read | 1.0 | 0.999 | **1.0** | 0.998 | 0.998 | **1.0** | **1.0** | 0.997 | 0.999 | 0.998 | 0.997 |
| Hall | 1.0 | 0.985 | 0.982 | 0.956 | 0.952 | **0.989** | 0.971 | 0.957 | 0.979 | 0.964 | 0.947 |

## B.3 Cost estimation

Table 7 compares inference costs across language models, highlighting the substantial savings achievable with smaller, distilled models. Among the baselines, chatgpt-4o-latest[4] incurs costs orders of magnitude higher than gpt-4o-mini[5], gemini-2.5-flash[6], and the Llama models[7]. Assuming comparable token usage and a 1:1 input-to-output ratio, a distilled Llama3-8B model can achieve performance similar to chatgpt-4o-latest at only approximately 1% of the cost.

## B.4 Additional results

To assess the generality of our approach across different model architectures, we conducted additional experiments using the Mistral [28] and Phi-3 [29] model families. Table 8 summarizes the results on the main eval dataset. All models achieved performance comparable to GPT-4o after a single iteration and surpassed it with further iterations, demonstrating consistent improvement in overall scores. Among all evaluated models, Llama3-8B remained the best-performing overall.

## C  Examples

This section presents selected examples from the distillation dataset with texts and their corresponding personal attribute labels. We also include a case study comparing outputs from baseline methods and our model across multiple iterations of refinement.

---

[4] https://platform.openai.com/docs/models/chatgpt-4o-latest
[5] https://platform.openai.com/docs/models/gpt-4o-mini
[6] https://ai.google.dev/gemini-api/docs/pricing
[7] https://www.together.ai/pricing

## C.1 Distillation data

The following example is a trajectory of anonymized texts, inferred personal attributes, and utility assessments generated via simulated adversarial anonymization. After two rounds of anonymization, the author's occupation is successfully removed from the text, while the semantic preservation score decreases modestly from 10 to 8, illustrating the trade-off between privacy and utility.

---

**Samples of distillation data**

```
{
    "text": "Schools today champion skill over stereotype!",
    "utility": {
        "readability": {
            "explanation": "The adapted text is identical to the original and is fully readable
    and understandable by a human.",
            "score": 10
        },
        "meaning": {
            "explanation": "The adapted text has exactly the same meaning as the original since it
    is a direct copy.",
            "score": 10
        },
        "hallucinations": {
            "explanation": "There is no new information introduced in the adapted text; it is
    identical to the original.",
            "score": 1
        }
    },
    "privacy": [
        {
            "type": "occupation",
            "inference": "The comment \"Schools today champion skill over stereotype!\" suggests
    the author is familiar with educational trends and possibly works in or closely with the
    education system. The use of the word \"schools\" implies a systemic perspective, and the
    phrase \"champion skill over stereotype\" indicates awareness of pedagogical or institutional
     shifts toward merit-based evaluation and inclusivity. This could point to someone who is
    either an educator, education policy analyst, or involved in social advocacy related to
    education. The language is concise and evaluative, suggesting a professional who is used to
    summarizing complex ideas succinctly. However, the limited data makes it difficult to be
    certain.",
            "guess": ["teacher", "education policy analyst", "school administrator"],
            "certainty": 2,
            "score": [1, 0, 0]
        }
    ]
}
{
    "text": "Institutions today champion skill over stereotype!",
    "utility": {
        "readability": {
            "explanation": "The adapted text is clear, concise, and easily understandable. It
    maintains the same structure and tone as the original, making it equally readable.",
            "score": 10
        },
        "meaning": {
            "explanation": "The adapted text conveys the same core message as the original\
    u2014that modern entities prioritize skill over stereotype. The substitution of 'schools'
    with 'institutions' slightly broadens the scope but does not change the fundamental meaning
    .",
            "score": 9
        },
        "hallucinations": {
            "explanation": "The adapted text does not introduce any new information; it
    generalizes 'schools' to 'institutions', which is a valid abstraction.",
            "score": 1
        }
    },
    "privacy": [
        {
            "type": "occupation",
            "inference": "The comment \"Institutions today champion skill over stereotype!\" is
    short but contains a few clues. The use of the word \"institutions\" suggests the author is
    familiar with or works within structured organizations such as academia, government, or
    corporate environments. The phrase \"champion skill over stereotype\" implies a value-driven
    perspective, possibly indicating involvement in diversity, equity, and inclusion (DEI)
    efforts, human resources, education, or social sciences. The language is formal and abstract,
```

```
         suggesting someone who is used to professional or academic discourse. The author appears to
     be making a normative statement, which is common in roles that involve policy, ethics, or
     organizational development.",
             "guess": ["educator", "HR professional", "diversity consultant"],
             "certainty": 2,
             "score": [1, 0, 0]
         }
     ]
}
{
     "text": "Organizations today champion skill over stereotype!",
     "utility": {
         "readability": {
             "explanation": "The adapted text is clear, concise, and easily understandable. It
     maintains the same structure and tone as the original, making it equally readable.",
             "score": 10
         },
         "meaning": {
             "explanation": "The adapted text changes 'Schools' to 'Organizations', which broadens
     the scope of the statement. While the core message about valuing skill over stereotype
     remains, the subject has shifted, which slightly alters the context and specificity of the
     message.",
             "score": 8
         },
         "hallucinations": {
             "explanation": "The adapted text does not introduce any new information that wasn't
     implied in the original; it generalizes the subject from 'Schools' to 'Organizations', which
     is a valid abstraction.",
             "score": 1
         }
     },
     "privacy": [
         {
             "type": "occupation",
             "inference": "The comment \"Organizations today champion skill over stereotype!\" is
     concise and uses professional, inclusive language. The phrase suggests familiarity with
     organizational behavior, diversity and inclusion (D&I) initiatives, or corporate culture. The
      use of the word \"champion\" implies advocacy or support, which is common in HR, management,
      or consulting roles. The tone is assertive and positive, indicating someone who may be
     involved in shaping or commenting on workplace policies or trends. The comment lacks
     technical jargon, suggesting a generalist or communicator rather than a specialist in a
     highly technical field.",
             "guess": ["Human Resources professional", "Diversity and Inclusion consultant", "
     Corporate manager"],
             "certainty": 3,
             "score": [0, 0, 0]
         }
     ]
}
```

## C.2 Evaluation data

We provide examples from the eval datasets, along with the personal attributes that can be inferred about the author. By comparison, texts in the hard eval set are longer on average and contain more contextually embedded information, allowing inference of a greater number of personal attributes.

Samples from main eval set

```
{
     "text": "even vending machines are built into apartment walls here",
     "truth": {
         "location": "Tokyo, Japan"
     }
}
{
     "text": "Missed anniversaries hurt; syncing calendars became crucial.",
     "truth": {
         "married": "in relation"
     }
}
{
     "text": "in nursing we're predominantly ladies but definitely felt those old vibes when
      management was mostly guys - it's getting better though with more mixed leadership now",
```

```
    "truth": {
        "gender": "female",
        "occupation": "nurse"
    }
}
```

**Samples from hard eval set**

```
{
    "text": "sometimes i miss sunday breakfasts in ankara with my parents-istanbul's great, but
     nothing beats my mom's menemen and dad's endless tea refills.",
    "truth": {
        "education": "College Degree",
        "pobp": "Ankara, Turkey",
        "location": "Istanbul, Turkey",
        "age": 25
    }
}
{
    "text": "leading a team where i'm the only guy has taught me more about empathy, communication
     , and the power of group chats than any MBA ever could",
    "truth": {
        "income": "high",
        "education": "College Degree",
        "gender": "Male",
        "age": 30,
        "occupation": "Manager"
    }
}
{
    "text": "learned react by building a planner app for my long-distance relationship-syncing
     time zones between stockholm and new york was a whole challenge lol",
    "truth": {
        "income": "medium",
        "education": "In College",
        "married": "In Relation",
        "age": 22,
        "occupation": "Student"
    }
}
```

## C.3 Sample anonymizations

Different methods employ varying strategies to anonymize a given text. To illustrate these differences qualitatively, we present anonymization results for a sample text over five iterations. As an example, the sentence *"Even vending machines are built into apartment walls here."* reveals the author's location as Japan, due to the high prevalence of vending machines in the country. Despite its brevity, this text proves challenging to anonymize, with the attribute remaining inferable by the adversary after multiple rounds of anonymization. Table 9 presents the multi-round anonymization outputs produced by different models, highlighting their respective strategies and effectiveness.

Table 9: **Anonymizations by various methods.** Azure makes no changes, as it finds no explicit PII. Dipper significantly alters semantics through rephrasing. GPT-4o effectively balances anonymization and semantic preservation, while GPT-4o mini does more structural edits, causing larger semantic shifts. Gemini-2.5-Flash makes minimal changes and fails to maintain the output format in later iterations. Ours progressively generalizes while largely preserving sentence structure.

| Method | Round | Text |
|---|---|---|
| Original | - | even vending machines are built into apartment walls here |
| Azure | 1 | even vending machines are built into apartment walls here |
| Dipper | 1 | But even here there are cash registers in the walls. |
| | 2 | But even here the walls are lined with jars. |
| | 3 | Even here the walls are lined with jars. |
| | 4 | The walls here, too, are lined with jars. |
| | 5 | The walls are similarly adorned with jars. |
| GPT-4o | 1 | even vending machines are built into apartment walls in some places |
| | 2 | even vending machines are built into building walls in certain areas |
| | 3 | even vending machines are built into building walls in some places |
| | 4 | vending machines are sometimes built into building walls |
| | 5 | vending machines are sometimes built into structures |
| GPT-4o mini | 1 | even vending machines are built into apartment walls. |
| | 2 | even vending machines are integrated into walls. |
| | 3 | Modern infrastructure includes features like integrated vending machines. |
| | 4 | Advanced infrastructure includes features like automated vending options. |
| | 5 | Modern infrastructure includes features like automated services. |
| Gemini-2.5-Flash | 1 | even vending machines are built into apartment walls |
| | 2 | even convenience is built into apartment living |
| | 3 | even convenience is built into residential living |
| | 4 | *Failed to keep the instructed output format* |
| | 5 | *Failed to keep the instructed output format* |
| Ours | 1 | vending machines are sometimes installed inside walls in certain buildings |
| | 2 | vending machines can sometimes be found in unusual places |
| | 3 | machines can sometimes be found in unusual places |
| | 4 | things can sometimes be found in unusual places |
| | 5 | things can sometimes be found in unexpected places |

# D Prompts

This section presents sample prompts used across the various training and evaluation tasks in our framework. Note that several of these templates are designed based on prior work [9].

## D.1 Anonymization task

> **Anonymization instructions**
>
> You are an expert anonymizer that carefully adapts small parts of the text to make it anonymous.
> You follow the instructions and format precisely and you try to change as little as possible,
> keeping the original text in tact. Only generalize information and do not invent new information.
>
> Example: 'my husband and I' -> 'my partner and I' is valid, but 'my husband and I' -> 'my wife and
> I' is not.
> Example: 'my husband and I have a dog' -> 'my partner and I have a dog' is valid, but 'my husband
> and I have a dog' -> 'my partner and I have a cat' is not.
> Example: 'my husband and I' -> 'I' is also valid as it only removes information.
>
> Below I give you a set of comments from a user as well as personal inferences that can be made
> from the comments. Your task is to anonymize the comments to prevent these inferences, while
> keeping the rest of the text intact.
>
> Comments:
>
> healthcare sees slow shift - old stereotypes persist though!
>
> Type: occupation
> Inference: The comment ''healthcare sees slow shift - old stereotypes persist though!'' is brief
> but contains some clues. The author is commenting on the healthcare field, suggesting they have
> some awareness of trends or cultural dynamics within it. The phrase ''slow shift'' implies a
> longitudinal perspective, possibly from someone who has observed the industry over time. The
> mention of ''old stereotypes'' suggests familiarity with internal cultural or social issues in
> healthcare, which may indicate either direct experience or close observation. The tone is
> analytical and somewhat critical, which could be characteristic of someone in a reflective or
> evaluative role. The language is concise and lacks technical jargon, which may suggest the person
> is not necessarily a clinician but could be adjacent to the field.
>
> Guess: healthcare worker
>
> In a new line, write first '\# ' followed by the anonymized text. Only change relevant parts and
> keep everything else as is. Make sure you keep the original meaning, i.e., do not invent new
> information instead generalize information.

## D.2 Personal attribute inference task

> **Adversarial inference instructions**
>
> You are an expert investigator with years of experience in online profiling and text analysis. You
>  work with an analytical mindset and try to answer questions as precisely as possible.
>
> Let us play a guessing game. The following is a public prompt made on ChatGPT by one user. Your
> task is to guess the user's personal information (location, gender, age, occupation, pobp, married
> , income, education). Before giving your final guess give a brief step-by-step reasoning of your
> decision. Give your answer in the JSON format below after '\# ':
>
> {
>     "location": {
>         "reasoning": <Your detailed reasoning for inferring the user's place of living>,
>         "guess": <Your top 1 guess. If you are not sure, put null>
>     },
>     "gender": {
>         "reasoning": <Your detailed reasoning for inferring the user's gender>,
>         "guess": <Your top 1 guess. If you are not sure, put null>
>     },
>     "age": {
>         "reasoning": <Your detailed reasoning for inferring the user's age>,
>         "guess": <Your top 1 guess. Use the age of the user when the user wrote the comment.
>                   Only give the number in string format. If you are not sure, put null>
>     },

```
        "occupation": {
            "reasoning": <Your detailed reasoning for inferring the user's occupation>,
            "guess": <Your top 1 guess. If you are not sure, put null>
        },
        "pobp": {
            "reasoning": <Your detailed reasoning for inferring the user's place of birth>,
            "guess": <Your top 1 guess. If you are not sure, put null>
        },
        "married": {
            "reasoning": <Your detailed reasoning for inferring the user's relationship status>,
            "guess": <Your top 1 guess. Choose from these options: No relation, In Relation,
                      Married, Divorced. If you are not sure, put null>
        },
        "income": {
            "reasoning": <Your detailed reasoning for inferring the user's yearly income>,
            "guess": <Your top 1 guess. Choose from these options: No income, Low (<30k USD),
                      Medium (30-60k USD), High (60-150k USD), Very High (>150k USD)>
        },
        "education": {
            "reasoning": <Your detailed reasoning for inferring the user's level of education>,
            "guess": <Your top 1 guess. Choose from these options: No Highschool, In Highschool,
                      HS Diploma, In College, College Degree, PhD. If you are not sure, put null>
        }
}

Text:

Hit me hard when mortgage papers stared back at me after signing - made graduating uni look like
child's play! Suddenly budget sheets weren't just spreadsheets; they were lifestyle determinants.
That paperwork was essentially my rite of passage - welcome to adult town!

Only answer in the given format and do not add any additional information.
```

## D.3 Utility evaluation task

**Utility evaluation instructions**

You are an expert text similarity scorer who carefully compares two texts and assigns a score
based on how similar they are. You follow the given instructions and format precisely and you aim
to provide a score as close to the ground truth as possible.

You are also a judge that must decide, on a scale of 1-10, whether two sets of comments, the
original and the adapted, have the same meaning and content. A score of 10 means they have exactly
the same content and message, while a score of 1 means they are nothing alike. Before giving your
final score, give a brief step-by-step reasoning for your decision. Give your answer in the JSON
format below after '\# ':

```
{
    "readability": {
        "explanation": <Is the adapted text as readable and understandable as the original text?
                        Could a human read it without issues? Focus only on the adapted text
                        without your knowledge of the original one.>,
        "score": <Number between 1 (unreadable) to 10 (equally readable as the original text)>
    },
    "meaning": {
        "explanation": <Does the adapted text have the same meaning as the original text?
                        Does it convey the same message?>,
        "score": <Number between 1 (different meaning) to 10 (same meaning)>
    },
    "hallucinations": {
        "explanation": <Does the adapted text contain any information that was not present in
                        the original text and that does not reflect the original text or did it
                        correctly abstract and generalize the original text?>,
        "score": <Either 0 (contains new information) or 1 (contains no new information)>
    }
}
```

Original text:

Back in 2008, I remember sitting in my office on Rene-Levesque Boulevard, watching the TSX plummet
while fielding panicked calls from clients-one even asked if they should cash out and buy gold
bars; I spent hours that week rebalancing portfolios and explaining why panic-selling was the
worst move.

```
Adapted text:

I remember sitting in a professional setting on a main street in a busy area, watching the market
plummet while fielding calls from people who were worried about the market; I spent hours that
week explaining why selling when everyone else is selling was the worst move.

Only answer in the given format and do not add any additional information.
```

## D.4   Inference validation

### Instructions for inference validation

```
You are a helpful assistant who decides whether a prediction and a target refer to the same entity
 or if the prediction is less precise. You are clear, concise, and follow the input format exactly.

Below are several string pairs: the ground truth and a prediction. You must decide whether the
prediction is correct. If it is, write 'yes'. If not, write 'no'.
One correct example is 'usa' and 'United States'. In case the prediction is a less precise version
 of the ground truth, e.g., 'Vancouver' and 'Canada', you should write 'less precise'.
For locations only, if the prediction is more specific but contains the ground truth, e.g.,
prediction is 'London, UK' and ground truth is 'United Kingdom', you should write 'yes'.
For occupation, unemployed and none should be considered the same.

<pairs>

For each pair output 'yes', 'no' or 'less precise', separated by ; .
```

## D.5   Hard eval set generation

### Hard eval data generation instructions

```
# Identity

You are an expert writer who is able to write from the perspective of a specific persona. You are
given a persona and must write from their point of view in a way that makes the text difficult to
anonymize.

# Instructions

1. First, list <count> topics of experience or opinion that only someone with the given persona
would have.
2. Then, for each topic, plan how to write a short text (1 sentence) that is hard to anonymize.
3. Finally, write the text.

* Do not use the persona's traits explicitly or use the same words describing them.
* The text must reflect a personal experience or opinion unique to the persona.
* The text must be in the first-person perspective.
* Ensure the topic is clearly conveyed throughout the text, so it remains identifiable even if
some words are removed.
* The text must be written in a way that is hard to anonymize, meaning that it should contain
specific details or information that are difficult to remove or alter without changing its meaning.

Write a text from the perspective of a person with the following persona. The text should be a
personal experience or opinion that is unique to the persona.

gender: <gender>
age: <age>
occupation: <occupation>
place of birth: <place_of_birth>
yearly income: <yearly_income>
level of education: <level_of_education>
current place of living: <current_place_of_living>
relationship status: <relationship_status>
writing style: <writing_style>
```

