# OpenReview forum: "Self-Refining Language Model Anonymizers via Adversarial Distillation"
_NeurIPS.cc/2025/Conference — NeurIPS 2025 poster_

### Official Review · Reviewer_hwtC · 2025-06-16

**Clarity:** 3
**Significance:** 3
**Originality:** 3
**Rating:** 4
**Confidence:** 3

**Summary:**

This paper proposes SEAL (Self-refining Anonymization with Language models), a novel distillation framework for training small language models (SLMs) to perform privacy-preserving text anonymization without relying on proprietary, external LLMs during inference. The approach leverages adversarial interactions between anonymizers and inference models (the same LLMs) to create trajectories of anonymized texts, attribute inferences, and utility scores. These are used to train SLMs via supervised fine-tuning (SFT) and direct preference optimization (DPO). The trained SLMs can iteratively self-refine the input text to achieve the privacy-utility trade-off. Experiments on the SynthPAI dataset show that SEAL-trained models (e.g., Llama3-8B) match or surpass the privacy-utility tradeoffs of GPT-4-based anonymizers.

**Questions:**

Q1. The authors are suggested to do more research on the privacy-utility tradeoff quantitatively.

Q2. The self-refinement loop seems automatic. Would it be feasible to guide it based on user preferences (For example, users are particularly sensitive to certain aspects of privacy like age)?

**Ethical Concerns:**

["NO or VERY MINOR ethics concerns only"]

**Final Justification:**

The design and implementation details have been clarified. I will maintain my positive score.

**Limitations:**

yes

**Quality:**

3

**Strengths And Weaknesses:**

Strengths:
1. The adversarial data generation method is innovative.
2. Cleverly introducing DPO through the setting of privacy and utility score functions has significantly improved performance on the basis of SFT.
3. The framework achieves iterative enhancement by simultaneously executing anonymization, adversarial inference, and utility evaluation in the same model.

Weaknesses:
1. There is no detailed explanation on how to obtain a confidence score.
2. In the setting, user should deploy the model locally for iteratively anonymization, the cost of this part is not provided.
3. The utility drops compared to baseline after anonymization.

---

> ### Author Rebuttal · Authors · 2025-07-31
>
> Dear Reviewer hwtC,
>
> We sincerely appreciate your review with thoughtful comments. We have carefully considered each of your questions and provide detailed responses below. Please let us know if you have any further questions or concerns.
>
> ---
>
> **[W1] How confidence scores are obtained**
>
> We thank the reviewer for pointing this out. We obtain the confidence scores by prompting the adversary model to assign a score to each inferred personal attribute on a scale from 1 to 5, where 1 indicates very low certainty (a pure guess with almost no evidence) and 5 indicates very high certainty (explicitly stated or undeniably inferrable). These confidence scores, along with the number of inferred attributes, are then used to compare the quality of different anonymizations.
>
> On a related note, while we initially suspected that such scores might be noisy, our ablation results (Section 5.3) show that incorporating model confidence leads to substantial improvements in training anonymizers. This suggests that confidence scores, when elicited from a sufficiently capable model, can provide consistent and informative signals that are valuable for learning.
>
> We will revise the draft to clarify these points.
>
> ---
>
> **[W2] Cost associated with deploying local models**
>
> We thank the reviewer for raising this important and practical question. While it is difficult to make a completely fair comparison (since GPT models are likely deployed on highly optimized infrastructure and we do not have access to similarly optimized hardware for on-device inference), we provide an approximate comparison based on 100 samples, assuming the use of faster inference APIs such as vLLM on a single NVIDIA A6000 GPU.
>
> The table below reports the mean and standard error of inference latency for (1) performing anonymization alone without adversarial feedback, and (2) performing adversarial inference followed by anonymization based on the results. Note that adversarial inference has notably higher latency than anonymization due to its longer output length. In the former case, the 8B model achieves slightly lower mean latency (0.939s vs. 1.094s) compared to GPT-4o. In the latter case, GPT-4o achieves lower latency (8.526s vs. 11.745s), though the 8B model’s latency arguably remains within a comparable range.
>
> \begin{array}{l|c|c}
> \hline
>  & \text{GPT-4o} & \text{Llama3-8B} \newline \hline
> \text{Anon. only} & 1.094 \pm 0.040 & 0.939 \pm 0.068 \newline
> \text{Anon. w/ adv. infer.} & 8.526 \pm 0.173 & 11.745 \pm 0.251 \newline\hline
> \end{array}
>
> We would also like to emphasize that a key motivation for leveraging small local models for anonymization is avoiding potential exposure of sensitive data to untrusted external systems. This offers significant privacy advantages, particularly in applications where sending raw user data to third-party APIs poses a high risk.
>
> ---
>
> **[W3,Q1] Further discussion on privacy-utility trade-offs**
>
> We thank the reviewer for raising this important point. Privacy and utility are inherently in a trade-off relationship, i.e., improving one often comes at the expense of the other. Hence, the key criterion in evaluating anonymization methods is the efficiency with which different methods achieve this trade-off.
>
> While the utility score drops slightly compared to adversarial anonymization (AA) with GPT-4o, the privacy gain is substantially larger, and the overall score, which is the difference between privacy gain and utility cost, is more than 20% higher than the baseline (0.253 vs. 0.305). Moreover, compared to AA with Gemini, our 8B model achieves strictly better results on both privacy and utility. Our model also yields a more desirable privacy-utility Pareto front than the baselines as illustrated in Figure 3. These results demonstrate that our method achieves efficient trade-offs and enables training SLM anonymizers that are competitive with or outperform frontier models.
>
> We will update the draft to provide a clearer explanation of these points.
>
> ---
>
> **[Q2] Guiding anonymization based on preferences**
>
> We appreciate the insightful question. There are several natural ways to adapt to different user preferences. One training-time approach is to adjust the comparison criteria during data curation by placing greater weight on specific aspects of privacy or utility that users care about. For example, instead of comparing two anonymizations solely based on the total number of inferred attributes, we can impose a higher weight on particular attributes such as age, encouraging the model to prefer anonymizations that better protect those attributes. Another approach, applicable at inference time, is to condition the model on user-specified privacy aspects, e.g., age or location, when performing anonymization, regardless of whether the model itself identifies them as inferable. This encourages the model to consistently account for the attributes users particularly want protected during the anonymization process.

---

> > ### Comment · Reviewer_hwtC · 2025-08-05
> >
> > Thanks for the authors' further clarification. I will maintain my positive score.

---

### Official Review · Reviewer_SaK9 · 2025-06-26

**Clarity:** 3
**Significance:** 2
**Originality:** 2
**Rating:** 4
**Confidence:** 4

**Summary:**

The authors introduce SElf-refining Anonymization with Language model (SEAL), a framework for distilling the anonymization capabilities of LLMs into small language models (SLMs), to do anonymization at inference time. They use adversarial interactions between an LLM anonymizer and an inference model to collect anonymized texts with inferred private attributes, which they use to train a SLM anonymizer via distillation. This is done using supervised fine-tuning (SFT) and direct preference optimization (DPO). They run experiments on SynthPAI, a dataset of synthetic personal profiles and text comments, to show that their method gives better privacy-utility tradeoffs than other baselines.

**Questions:**

- How is the hard eval dataset constructed? It would be useful to give more details about this dataset since it is new to this paper.
- How does SEAL perform on other types of datasets?

**Ethical Concerns:**

["NO or VERY MINOR ethics concerns only"]

**Final Justification:**

My concerns regarding the limited evaluation have been addressed, hence I am raising my score.

**Limitations:**

yes

**Paper Formatting Concerns:**

None.

**Quality:**

3

**Strengths And Weaknesses:**

Strengths
- The experimental results show that SEAL gives better privacy than the other baselines, for the dataset used.
- The authors compare the performance against a wide range of baselines, and also include evaluations against a hard dataset with contextually embedded personal information.
- The authors also present privacy-utility tradeoffs when using SEAL across different models with a range of sizes.

Weaknesses
- Only synthetic data is used for the evaluation, which may differ substantially from real data.
- It is unclear if the results generalize to different types of data, since only one type of dataset was used. Furthermore, the dataset seems to consist of short comments, and it is unclear if SEAL works with longer text common in other applications that require privacy.
- The evaluation of privacy and utility are done using GPT-4. This means that the results on privacy are dependent on what GPT-4 is able to detect, which may differ from actual humans, and similarly for the utility results.

---

> ### Author Rebuttal · Authors · 2025-07-31
>
> Dear Reviewer SaK9,
>
> We sincerely appreciate your review with thoughtful comments. We have carefully considered each of your questions and provide detailed responses below. Please let us know if you have any further questions or concerns.
>
> ---
>
> **[W1] Use of synthetic data**
>
> We thank the reviewer for raising this important point. We acknowledge the potential limitation of using synthetic data. However, this is an inherent challenge in personal attribute inference (PAI) research, where the use of real user data poses significant ethical and privacy concerns, i.e., hence, is nearly impossible. To this end, we used the SynthPAI dataset [1], which consists of user profiles and comments generated using GPT-4 but carefully designed to resemble real data. In [1], the authors conduct a human study with 40 participants, showing that humans perform near randomly in distinguishing synthetic comments from real Reddit comments. They also demonstrate that the user profiles and comments are highly distinct from one another. We believe these results support the validity of our experimental setup and that our evaluation on a distinct set of user profiles provides a reasonable assessment of generalization performance. Nevertheless, developing more diverse and comprehensive datasets remains an important direction for future PAI research.
>
> We will revise the paper to provide a clearer explanation of these points.
>
> ---
>
> **[W2, Q2] Generalization to additional data**
>
> We thank the reviewer for this important comment. We first briefly reiterate key points about SynthPAI from our response to W1 above. The authors, in an extensive human study, found that the user profiles and comments in the dataset are highly distinct from one another. They also found that the synthetic comments were nearly indistinguishable from real ones to human evaluators. We believe these characteristics allow us to perform a reasonable evaluation of generalization performance.
>
> For further assessment, we constructed an additional evaluation dataset specifically designed to include texts in which private information is more deeply embedded in context (please refer to our response to Q1 below for details on its construction). As summarized in Table 2, our empirical results show that the 8B model, trained solely on the original SynthPAI training split, performed competitively against frontier models, demonstrating its generalization capabilities. Nevertheless, we acknowledge that developing more diverse PAI benchmarks remains an important future research direction and would further strengthen empirical validation.
>
> Lastly, while SynthPAI primarily consists of text comments typically found on social media such as Reddit, which do tend to be relatively short, we provide several longer examples generated using the 8B model.
>
> **Example 1.** The model anonymizes a personal anecdote by generalizing “ran a workshop” to “was involved in a situation” and removing role-specific terms like “junior dev” and “senior engineers.” References to “stereotypical nerd” are also omitted. These edits preserve the story’s core message while reducing occupational and stylistic identifiers.
> - Original: “Mighty interwebs throw curveballs sometimes – ran a workshop where everyone assumed the junior dev was lead because \"stereotypical nerd\" profile didn't fit their view of senior engineers! Minds blown when they realized who called the shots 😏”
> - Anonymized: “Sometimes the internet throws curveballs – was involved in a situation where everyone assumed a certain person was in charge because they didn't fit the typical image of someone in a leadership role! Minds blown when it turned out who was actually calling the shots”
>
> **Example 2.** For this example, the model abstracts financial and entrepreneurial specifics. “All-in on stocks” becomes “Thinking about investments,” and “small biz” is generalized to “local opportunities.” The rewritten version softens personal claims and removes stylistic markers like “gold mines” and “savvy,” while maintaining the original message about diverse investment opportunities.
> - Original: “All-in on stocks? Don't forget small biz can be gold mines too! My own experience says local opportunities can really pay off if you're savvy enough – just gotta keep your ear close to the ground!”
> - Anonymized: “Thinking about investments? Don't forget local opportunities can be valuable too! My own experience says being open to different possibilities can really pay off if you're willing to try – just gotta stay informed!”
>
> **Example 3.** The model generalizes both occupation and tone. “Running plates n' nights crunching numbers” becomes “doing different tasks and analyzing things,” removing occupational clues and informal phrasing. The expression “flips your mindset” is softened to “changes how you think,” while maintaining the core idea about developing an efficiency-oriented mindset.
> - Original: “spending days running plates n' nights crunching numbers totally flips your mindset - like you start optimizing everything...got me thinking efficiency 24/7 😅”
> - Anonymized: “spending time doing different tasks and analyzing things really changes how you think – like you start trying to make everything better...got me thinking about how everything can be improved.”
>
> ---
>
> **[W3] Use of GPT-4 as an evaluator**
>
> We thank the reviewer for raising this important point and the opportunity to clarify.
>
> First, we note that prior work has shown GPT-4 to be among the strongest available models for personal attribute inference (PAI), achieving approximately 80% of human performance on SynthPAI [1]. Our results, where we use an even more recent version of GPT-4, therefore demonstrate that our method enables training of SLM anonymizers that are effective against one of the most capable adversaries currently available.
>
> Secondly, to further assess performance with different LLM judges, we evaluated our models distilled from GPT-4o using Claude Sonnet 4 and Gemini 2.5 Flash. While absolute scores vary slightly, the overall trend remains consistent: our 8B model outperforms GPT-4o in both privacy and overall scores across evaluators, with further gains from additional rounds of self-refinement. Specifically, with Claude as the judge, our 8B model achieves an overall score of 0.371 compared to GPT-4o’s 0.262. The improvement is even more substantial with Gemini, where our model scores 0.403 compared to GPT-4o’s 0.256. We observed that, on the eval dataset, Claude tended to assign slightly stricter utility scores compared to GPT-4.1 and Gemini.
>
> \begin{array}{l|c|c|ccc}
> \hline
> \text{Claude} & \text{Original} & \text{GPT-4o} &  & \text{Llama3-8B} &  \newline
>  &  &  & \text{iter1} & \text{iter2} & \text{iter3} \newline\hline
> \text{Overall} \uparrow & - & 0.262 & 0.371 & \underline{0.416} & \textbf{0.447} \newline\hline
> \text{Privacy} \downarrow & 0.716 & 0.495 & 0.389 & \underline{0.310} & \textbf{0.251} \newline
> \text{Utility} \uparrow & 0.999 & \textbf{0.952} & \underline{0.913} & 0.848 & 0.796 \newline\hline
> \end{array}
>
> \begin{array}{l|c|c|ccc}
> \hline
> \text{Gemini} & \text{Original} & \text{GPT-4o} &  & \text{Llama3-8B} &  \newline
>  &  &  & \text{iter1} & \text{iter2} & \text{iter3} \newline\hline
> \text{Overall} \uparrow & - & 0.256 & 0.403 & \underline{0.490} & \textbf{0.493} \newline\hline
> \text{Privacy} \downarrow & 0.669 & 0.461 & 0.348 & \underline{0.241} & \textbf{0.202} \newline
> \text{Utility} \uparrow & 1.0 & \textbf{0.945} & \underline{0.923} & 0.850 & 0.795 \newline\hline
> \end{array}
>
> Lastly, to further examine how well GPT-4 aligns with human judgment in evaluating utility, we conducted a human study with three participants who rated 100 samples according to the following criteria:
> - Readability: How readable the adapted text is compared to the original, rated on a scale from 1 to 10
> - Semantic preservation: How well the original semantics are retained, also rated on a scale from 1 to 10
> - Hallucination: Whether the adapted text introduces any new information not present in the original, rated as a binary 0 or 1
>
> We report average Pearson correlation coefficients for readability and semantic preservation, and accuracy for hallucination, using human annotations as ground-truth. For readability, which is more subjective, we obtained an average correlation of **0.717**, and for semantic preservation, a substantially higher correlation of **0.814**. For hallucination, GPT-4 achieves an average accuracy of **0.775** relative to human annotations. These results indicate strong alignment between GPT-4  and human evaluations across all three utility criteria.
>
> We will revise our manuscript to include these additional results.
>
> ---
>
> **[Q1] Construction of the hard eval dataset**
>
> We thank the reviewer for the question. While we provide the details in the appendix, an explicit reference in the main text would enhance clarity. Briefly, the dataset is specifically designed to contain texts with private information more deeply embedded in context. We first identify samples from SynthPAI that are particularly difficult to anonymize using multiple iterations of adversarial anonymization with GPT-4. Based on qualitative analysis of these examples, where PII is revealed through contextual and narrative cues rather than explicit identifiers, we generate 500 new texts from held-out profiles, guiding the model to emulate these characteristics. Applying the same anonymization process shows that a significantly larger proportion of this dataset remains unanonymized (see Figure 5), demonstrating its greater difficulty compared to the main eval data from SynthPAI.
>
> We will revise the draft to clarify how the hard eval dataset is constructed and how it differs from the main eval set.
>
> ---
>
> **References**
>
> [1] Yukhymenko et al. “A Synthetic Dataset for Personal Attribute Inference.” NeurIPS, 2024.

---

> > ### Comment · Reviewer_SaK9 · 2025-08-04
> >
> > Thank you for the clarifications and the additional evaluations. I have raised my score accordingly.

---

> > > ### Author Response · Authors · 2025-08-05
> > >
> > > Dear Reviewer SaK9,
> > >
> > > We sincerely appreciate your thoughtful response and your re-evaluation of the paper. We will fully incorporate the additional results and clarifications based on your feedback to further strengthen the work.
> > >
> > > Thank you again for your constructive feedback and for increasing your rating.
> > >
> > > Authors

---

### Official Review · Reviewer_6CMa · 2025-07-01

**Clarity:** 2
**Significance:** 3
**Originality:** 3
**Rating:** 5
**Confidence:** 3

**Summary:**

This paper introduces SElf-refining Anonymization with Language model (SEAL), a novel framework to distill the capabilities of large, proprietary models into smaller, locally-run language models (SLMs) for text anonymization. The method uses data from adversarial LLM interactions to train an SLM to both anonymize text and critique its own outputs for privacy and utility. This enables the SLM to iteratively improve its anonymizations via self-refinement at inference time without external model feedback. Experiments show that an 8B model trained with SEAL can match or even surpass the privacy-utility tradeoff of a GPT-4-based anonymizer

**Questions:**

Questions:

1. Can the core principle of distilling generation and critique capabilities for self-refinement be generalized to other complex tasks, such as summarization or code generation, where a model could iteratively refine its output by checking for coherence, correctness, or logical consistency?

2. What is the practical inference latency for multi-step self-refinement on an 8B model, and how does this compare to the latency of a single API call to a larger model like GPT-4o?

3. Table 1 shows that after one iteration, SEAL (8B) has a slightly lower utility score (0.931) than the GPT-4o baseline (0.947). However, after three iterations, SEAL's privacy is significantly better, while utility drops to 0.862. Could you comment on this privacy-utility tradeoff? Is it possible to tune the model or the self-refinement process to prioritize utility preservation while still achieving substantial privacy gains?

**Ethical Concerns:**

["NO or VERY MINOR ethics concerns only"]

**Limitations:**

yes

**Quality:**

3

**Strengths And Weaknesses:**

Strengths:

The paper addresses the critical need for private, efficient, and accessible anonymization tools. The proposed self-refining distillation framework, which uniquely teaches a model to critique its own outputs, is a significant and original contribution.

The methodology is sound, and the experimental evaluation is comprehensive. The use of strong baselines, varied datasets, and detailed ablation studies provides convincing evidence for the framework's effectiveness.

The paper is exceptionally well-written and clearly structured, with effective figures that make the complex framework easy to understand.

Weaknesses:

The practical latency of multi-step self-refinement is not discussed, leaving a gap in understanding the full efficiency trade-off compared to single API calls.

The privacy evaluation is benchmarked against a specific model (GPT-4), meaning the anonymization is not guaranteed to hold against future, more capable adversaries.

The distilled critic model may be over-reliant on the style of its teacher model (GPT-4o), potentially limiting its effectiveness on out-of-distribution text.

---

> ### Author Rebuttal · Authors · 2025-07-31
>
> Dear Reviewer 6CMa,
>
> We sincerely appreciate your review with thoughtful comments. We have carefully considered each of your questions and provide detailed responses below. Please let us know if you have any further questions or concerns.
>
> ---
>
> **[W1,Q2] Latency comparison with respect to API calls**
>
> We thank the reviewer for raising this important and practical question. While it is difficult to make a completely fair comparison (since GPT models are likely deployed on highly optimized infrastructure and we do not have access to similarly optimized hardware for on-device inference), we provide an approximate comparison based on 100 samples, assuming the use of faster inference APIs such as vLLM on a single NVIDIA A6000 GPU.
>
> The table below reports the mean and standard error of inference latency for (1) performing anonymization alone without adversarial feedback, and (2) performing adversarial inference followed by anonymization based on the results. Note that adversarial inference has notably higher latency than anonymization due to its longer output length. In the former case, the 8B model achieves slightly lower mean latency (0.939s vs. 1.094s) compared to GPT-4o. In the latter case, GPT-4o achieves lower latency (8.526s vs. 11.745s), though the 8B model’s latency arguably remains within a comparable range.
>
> \begin{array}{l|c|c}
> \hline
>  & \text{GPT-4o} & \text{Llama3-8B} \newline \hline
> \text{Anon. only} & 1.094 \pm 0.040 & 0.939 \pm 0.068 \newline
> \text{Anon. w/ adv. infer.} & 8.526 \pm 0.173 & 11.745 \pm 0.251 \newline\hline
> \end{array}
>
> We would also like to emphasize that a key motivation for leveraging small local models for anonymization is avoiding potential exposure of sensitive data to untrusted external systems. This offers significant privacy advantages, particularly in applications where sending raw user data to third-party APIs poses a high risk.
>
> ---
>
> **[W2] Anonymization against more capable adversaries**
>
> We thank the reviewer for raising this important point and fully agree that privacy evaluations should consider the evolving capabilities of adversarial models.
>
> First, we note that prior work has shown GPT-4 to be among the strongest available models for personal attribute inference, achieving approximately 80% of human performance on SynthPAI [1]. Our results, where we use an even more recent version of GPT-4, therefore demonstrate that our method enables training of SLM anonymizers that are effective against one of the most capable adversaries currently available.
>
> More importantly, our framework is designed to be adaptable. Since the distillation data is collected from the adversary model itself, it can be naturally extended to future, more capable adversaries. As more powerful models become available, the same procedure can be applied to gather updated distillation data and train anonymizers that remain robust in more challenging settings.
>
> ---
>
> **[W3] Generalization to out-of-distribution texts**
>
> We thank the reviewer for this important comment. We agree that a potential limitation of distilling from a single teacher model is that the model may overfit to the teacher’s stylistic patterns.
>
> That said, we would like to note that the authors of the SynthPAI dataset conducted a human study with 40 participants, demonstrating that the user profiles and comments are highly distinct from one another [1]. They also found that humans perform near chance in distinguishing synthetic comments from real Reddit comments. Also, while we distill from the more recent GPT-4o, we note that the SynthPAI dataset was generated using an earlier version of GPT-4 (gpt-4-0613, released in June 2023). We believe these results support that our empirical setup of evaluating on a distinct set of user profiles provides a reasonable measure of generalization performance. Nevertheless, we acknowledge that evaluation on more diverse settings would offer deeper insights into the generalizability of the distilled models, and we consider this an important direction for future work.
>
> ---
>
> **[Q1] Application of the framework to other domains**
>
> We thank the reviewer for this insightful comment and fully agree that the core principle of distilling generation and critique capabilities for self-refinement is broadly applicable beyond anonymization. Specifically, our approach of collecting data trajectories and using a two-step training process, i.e., task adaptation followed by preference learning, to train student models to generate diverse yet high-quality outputs is naturally applicable to a wider range of domains. The framework is especially well-suited to tasks like summarization and code generation, where iterative refinement and the need to balance multiple objectives (e.g., coherence and correctness) are both common and essential. Several key findings from our application to anonymization are likely transferable to such domains, including the value of training to self-critique, the use of model’s own evaluation during refinement, and the utility of leveraging auxiliary signals such as confidence scores from the teacher models.
>
> We will revise the draft to incorporate this discussion and motivate broader application of our framework to relevant tasks.
>
> ---
>
> **[Q2] Further discussion on privacy-utility trade-offs**
>
> We thank the reviewer for raising this important point. Privacy and utility are inherently in a trade-off relationship, i.e., improving one often comes at the expense of the other. Hence, the key criterion in evaluating anonymization methods is the efficiency with which different methods achieve this trade-off.
>
> While the utility score drops slightly compared to adversarial anonymization (AA) with GPT-4o, the privacy gain is substantially larger, and the overall score, which is the difference between privacy gain and utility cost, is more than 20% higher than the baseline (0.253 vs. 0.305). Moreover, compared to AA with Gemini, our 8B model achieves strictly better results on both privacy and utility. Our model also yields a more desirable privacy-utility Pareto front than the baselines as illustrated in Figure 3. These results demonstrate that our method achieves efficient trade-offs and enables training SLM anonymizers that are competitive with or outperform frontier models.
>
> Prioritizing utility preservation is also straightforward within our framework. One natural approach is to adjust the comparison criteria during data curation by placing a higher weight on utility. For example, instead of preferring $s_i$ over $s_j$ when $p(s_i) > p(s_j)$ and $u(s_i) \geq u(s_j)$, we can impose a stricter utility margin, requiring $u(s_i) > u(s_j) + \delta_u$ for some $\delta_u > 0$. This encourages the model to favor anonymizations that better preserve utility while still improving privacy.
>
> We will update the draft to provide a clearer explanation of these points.
>
> ---
>
> **References**
>
> [1] Yukhymenko et al. “A Synthetic Dataset for Personal Attribute Inference.” NeurIPS, 2024.

---

> > ### Comment · Reviewer_6CMa · 2025-08-06
> >
> > Thanks for the detailed responses. I will maintain my original score.

---

### Official Review · Reviewer_wVoE · 2025-07-02

**Clarity:** 3
**Significance:** 2
**Originality:** 2
**Rating:** 4
**Confidence:** 4

**Summary:**

In this work, authors tackle the task of text anonymization to remove the PII information. To this end, authors curate a high-quality dataset of input-output pairs of data by using SOTA adversarial anonymization. Next, authors use distillation to impart the anonymization capabilities to SLMs via two steps, SFT followed by DPO. Authors validate the effectiveness of the distilled SLM on the SynthPAI dataset and showcase the versatility of their approach.

**Questions:**

- Please see weakness on human validation of results, prior works, experimental setting.

-  The prompt used in inference time for the step discussed in Line 160-161 is not clear. This seems to differ from the training time prompt style, where it takes text $x_i$ and outputs either $x_j$, $P$, $U$. But at the inference, both privacy and utility are conditioned, which differs from training prompts. Please provide more details to this.

-  I have concerns on the experimental setting relying on GPT-4 both for dataset traces, and also using during evaluation stage. Authors could discuss the generalization of their training dataset construction with other models.

-   The performance of the anonymizer is dependent on the architecture family. SEAL Qwen-3-8B is on par with the performance of AA with larger models. More experiments with other families will be helpful as the current  training of SLMs is very minimal with <2 epochs of SFT and DPO.

- More qualitative examples showcasing the differences with AA [9] wrt to each attribute is needed to understand the quality. While paper provides objective metrics, it would benefit the reader to highlight striking differences to baselines in terms of anonymization, along with evaluation (privacy and utility) metrics.

**Ethical Concerns:**

["NO or VERY MINOR ethics concerns only"]

**Final Justification:**

My concerns regarding the evaluation with LLM-as-judge framework, differences with prior work is addressed. I increase my rating to BA.

**Limitations:**

yes

**Quality:**

2

**Strengths And Weaknesses:**

Strengths
- The proposed method is particular effective in the real-world setting as the entire anonymization happens on device without any dependency on the big models
- The method is easy to follow and clearly explained
- Authors also contribute to a new evaluation set of 600+ datapoints to assess the anonymization capabilities

Weakness
- Technical novelty of the paper is limited and shares the similar theme with prior works. Already works such as [a, b] show that text anonymization capabilities can be imparted to SLMs by distillation and work of [a] uses a very similar methodology of adversarial anonymization of [9] to curate distillation dataset.

- LLM-as-a-judge is used for both privacy and utility evaluation. The results especially for utility evaluation needs human validation to understand how reliable the evaluation metrics with GPT-4 and their correlation to human evaluation.

- Authors use GPT-4o for dataset generation, and also use GPT-4 for offline evaluation of text anonymization. I am concerned of the bias introduced due to this setting as there are recent works which show Judge has preference to their own generations [c].

[a] IncogniText: Privacy-enhancing Conditional Text Anonymization via LLM-based Private Attribute Randomization, SafeGenAI workshop, NeurIPS 2024

[b] The Fire Thief Is Also the Keeper: Balancing Usability and Privacy in Prompts, 2024

[c] LLM Evaluators Recognize and Favor Their Own Generations, 2024

---

> ### Author Rebuttal · Authors · 2025-07-31
>
> Dear Reviewer wVoE,
>
> We sincerely appreciate your review with thoughtful comments. We have carefully considered each of your questions and provide detailed responses below. Please let us know if you have any further questions or concerns.
>
> ---
>
> **[W1,Q1] Comparisons with prior work**
>
> We thank the reviewer for the feedback and the opportunity to clarify the technical novelty of our work. While prior methods such as IncogniText [a] and ProSan [b] address privacy-preserving text anonymization, SEAL introduces a distinct and technically novel framework that:
> 1) distills not only anonymization but also adversarial inference and utility evaluation into a single model, enabling iterative self-refinement;
> 2) employs a two-step training process, i.e., task adaptation followed by preference learning, to train student models that achieve superior privacy-utility trade-offs; and
> 3) enables the training of SLM anonymizers that are competitive with or outperform frontier models in anonymization quality, while maintaining strong utility.
>
> These contributions go far beyond prior work, which relies on simple instruction fine-tuning for anonymization alone, does not explicitly optimize privacy-utility trade-offs when training student models, or lacks empirical validation of anonymizer effectiveness against frontier models.
>
> To further elaborate:
> - **Distillation of anonymization and critique capabilities into a single model**:
> SEAL trains a single model that can both anonymize text and critique its own outputs. It learns to infer private attributes and evaluate utility (e.g., readability and semantic preservation), enabling iterative self-refinement without requiring any external feedback at inference time. By distilling from a target adversary, SEAL produces SLM anonymizers that remain effective even against advanced adversarial models. In contrast, IncogniText distills only the anonymization capability, and thus either relies on external models for feedback or on the local model itself—which, without explicit fine-tuning for adversarial inference or utility evaluation, is insufficient for effective self-refinement (see Table 3 [a]).
> - **Explicit optimization of privacy-utility trade-off in fine-tuning**:
> SEAL performs supervised fine-tuning on anonymization and critique tasks, followed by preference learning to explicitly bias generation toward better privacy-utility trade-offs. This allows training SLM anonymizers that achieve superior trade-offs even compared to frontier models. In contrast, IncogniText uses simple supervised fine-tuning for anonymization alone, resulting in modest privacy gains but significantly reduced utility, as shown in Table 3 [a]. Also, ProSan focuses on per-token perturbation heuristics without optimizing end-to-end trade-offs.
> - **Empirical validation against frontier models**:
> Our experiments show that an 8B model trained with SEAL can outperform GPT-4o in anonymization quality while maintaining strong utility. Neither framework has empirically demonstrated a similar level of performance against frontier models.
>
> ---
>
> **[W2] Human evaluation and LLM-as-a-Judge**
>
> We thank the reviewer for the opportunity to clarify.
>
> First, we note that prior work has shown GPT-4 to be among the strongest available models for personal attribute inference (PAI), achieving approximately 80% of human performance on SynthPAI [1]. Our results, where we use an even more recent version of GPT-4, therefore demonstrate that our method enables training of SLM anonymizers that are effective against one of the most capable adversaries currently available. Please refer to our responses to W3 and Q3 below for results with other frontier models as LLM judges.
>
> To further examine how well GPT-4 aligns with human judgment in evaluating utility, we conducted a human study with three participants who rated 100 samples according to the following criteria:
> - Readability: How readable the adapted text is compared to the original, rated on a scale from 1 to 10
> - Semantic preservation: How well the original semantics are retained, also rated on a scale from 1 to 10
> - Hallucination: Whether the adapted text introduces any new information not present in the original, rated as a binary 0 or 1
>
> We report average Pearson correlation coefficients for readability and semantic preservation, and accuracy for hallucination, using human annotations as ground-truth. For readability, which is more subjective, we obtained an average correlation of **0.717**, and for semantic preservation, a substantially higher correlation of **0.814**. For hallucination, GPT-4 achieves an average accuracy of **0.775** relative to human annotations. These results indicate strong alignment with human evaluations across all three criteria.
>
> ---
>
> **[W3,Q3] Use of GPT-4 models for data generation and evaluation**
>
> We appreciate the reviewer’s point. We would first like to clarify that the versions of GPT-4 used for data generation and evaluation are, in fact, different. For collecting distillation data, we used GPT-4o, while for evaluation, we used GPT-4.1, a more recent version of GPT-4.
>
> We also evaluated our model distilled from GPT-4o using Claude Sonnet 4 and Gemini 2.5 Flash. While absolute scores vary slightly, the overall trend remains consistent: our 8B model outperforms GPT-4o in both privacy and overall scores, with further gains from self-refinement. Specifically, for Claude, our 8B model achieves an overall score of 0.371 compared to GPT-4o’s 0.262. The improvement is even more substantial with Gemini, where our model scores 0.403 compared to GPT-4o’s 0.256. We observed that Claude tended to assign slightly stricter utility scores compared to GPT-4.1 and Gemini.
>
> \begin{array}{l|c|c|ccc}
> \hline
> \text{Claude} & \text{Original} & \text{GPT-4o} &  & \text{Llama3-8B} &  \newline
>  &  &  & \text{iter1} & \text{iter2} & \text{iter3} \newline\hline
> \text{Overall} \uparrow & - & 0.262 & 0.371 & \underline{0.416} & \textbf{0.447} \newline\hline
> \text{Privacy} \downarrow & 0.716 & 0.495 & 0.389 & \underline{0.310} & \textbf{0.251} \newline
> \text{Utility} \uparrow & 0.999 & \textbf{0.952} & \underline{0.913} & 0.848 & 0.796 \newline\hline
> \end{array}
>
> \begin{array}{l|c|c|ccc}
> \hline
> \text{Gemini} & \text{Original} & \text{GPT-4o} &  & \text{Llama3-8B} &  \newline
>  &  &  & \text{iter1} & \text{iter2} & \text{iter3} \newline\hline
> \text{Overall} \uparrow & - & 0.256 & 0.403 & \underline{0.490} & \textbf{0.493} \newline\hline
> \text{Privacy} \downarrow & 0.669 & 0.461 & 0.348 & \underline{0.241} & \textbf{0.202} \newline
> \text{Utility} \uparrow & 1.0 & \textbf{0.945} & \underline{0.923} & 0.850 & 0.795 \newline\hline
> \end{array}
>
> ---
>
> **[Q2] Clarification of the prompt used**
>
> We appreciate the reviewer’s question. It is important to note that the anonymization of input text $x_i$ is conditioned on the set of private attributes that an adversary might infer from $x_i$, allowing the model to account for potential privacy risks during generation. At training time, the model learns to anonymize texts conditioned on such evaluations provided by a teacher model (i.e., the adversary we distill from). At inference time, the model instead performs its own evaluation—learned during training—to identify potentially inferable attributes and uses this self-evaluation to guide the anonymization. To assess the impact of this conditioning, we conducted an ablation study and observed substantial improvements in anonymization performance (see the “Adv. Feed.” column in Table 3).
>
> ---
>
> **[Q4] Experiments with other model families**
>
> We appreciate the suggestion. To further evaluate our method on alternative model architectures, we conducted additional experiments using the Mistral and Phi-3 models. The table below summarizes the results on the main evaluation dataset. Overall, all models achieve performance comparable to GPT-4o after a single iteration and surpass it with subsequent iterations, with overall scores consistently improving. Among all the models we experimented with, Llama3-8B was the best performing overall.
>
> \begin{array}{l|c|ccc|ccc|ccc}
> \hline
> & \text{GPT-4o} & & \text{Mistral-7B} &  & & \text{Ministral-8B} &  &  & \text{Phi-3-medium} &  & \newline
>  &  & \text{iter1} & \text{iter2} & \text{iter3} & \text{iter1} & \text{iter2} & \text{iter3} & \text{iter1} & \text{iter2} & \text{iter3} \newline\hline
> \text{Overall} \uparrow & 0.253 & 0.247 & 0.301 & 0.374 & 0.242 & 0.314 & 0.351 & 0.245 & 0.320 & 0.397 \newline\hline
> \text{Privacy} \downarrow & 0.434 & 0.436 & 0.387 & 0.334 & 0.443 & 0.385 & 0.352 & 0.438 & 0.376 & 0.311 \newline
> \text{Utility} \uparrow & 0.947 & 0.945 & 0.920 & 0.908 & 0.951 & 0.930 & 0.914 & 0.946 & 0.922 & 0.895 \newline\hline
> \end{array}
>
> ---
>
> **[Q5] More qualitative examples**
>
> We appreciate the helpful suggestion. We agree that including more examples would better illustrate the qualitative differences among anonymization methods. Due to space constraints, we only provide a single example here but will include more in our final draft.
>
> **Example.** In this example, the sensitive attribute is the author’s status as a business owner. The original text contains the phrase *Running your own biz*, which directly reveals this information. While Gemini-2.5-Flash removes the phrase entirely, potentially over-sacrificing utility, GPT-4o and GPT-4o-mini replace it with more general references to *responsibilities*. Our 8B model offers a more targeted rewrite, i.e., *Working independently*, which preserves the original sentence’s intent while obfuscating the specific occupational detail.
> - Original: “Running your own biz really puts things into perspective!”
> - Ours: “Working independently really puts things into perspective!”
> - GPT-4o: “Taking on new responsibilities really puts things into perspective!”
> - GPT-4o-mini: “Having your own responsibilities really puts things into perspective!”
> - Gemini-2.5-Flash: “This really puts things into perspective!”

---

> > ### Comment · Reviewer_wVoE · 2025-08-04
> >
> > Dear Authors,
> >
> > Thank you for the rebuttal experiments. My concerns are largely addressed.
> >
> > [W1Q1] Thank you for clarifying the contributions clearly. I agree with your point that prior works only distilled the anonymization capabilities but does not distill critique capabilities. Please refine the discussion in the revised paper and appropriately position the prior work.
> >
> > [W2] Thank you for conducting the human study. The correlation coefficients are reasonable but could be further investigated to understand if there are systemic issues in the anonymizer. Please incorporate this into the revised paper.
> >
> > [W3] Please add this section with results using different judges for evaluation. This is be valuable for the community in the future to not limit the evaluations to one single model.
> >
> > [Q2] Sorry, after reading the paper again. I  have realised my confusion. The dataset D_priv used in the finetuning imparts the critique capabilities to the SLM. May I suggest to add an additional algorithm (like the Algorithm 1 in main paper) in the Appendix with inference time logic iterating over inferring the private attributes, computing utility and then self-refining.
> >
> > I believe the proposed algorithm is intuitive and efficiently distils the  different capabilities of LLMs into SLMs. However, at the core, the technical novelty is incremental to the prior work but the paper provides a stronger evidence generalising over multiple attributes, multiple SLM backbones and also optimises the utility metrics.
> >
> > I increase the rating to WA and request authors to incorporate all additional experiments into the revised version.

---

> > > ### Author Response · Authors · 2025-08-04
> > >
> > > Dear Reviewer wVoE,
> > >
> > > We sincerely appreciate your thoughtful response and for re-evaluating the paper. We’re glad to hear that the additional experiments and clarifications addressed your concerns. We will revise the paper to clarify the distinction between SEAL and prior work, especially around the integration of critique capabilities, and will incorporate the human study results and evaluations using multiple judges as suggested. We also appreciate the suggestion to include a detailed algorithm outlining the inference-time self-refinement process, which we will add to the appendix.
> > >
> > > We’re encouraged by your recognition that our approach generalizes across attributes and model families while optimizing utility. We believe that incorporating the additional results and clarifications will further strengthen our paper.
> > >
> > > Thank you again for your constructive feedback and for increasing your rating.
> > >
> > > Authors

---

### Comment · Area_Chair_bixN · 2025-08-03
**Discussion period**

Dear Reviewers,

Thank you so much for all your time and effort supporting NeurIPS!

If you haven't yet, please take a moment to read through the author's rebuttal. If the rebuttal addresses your concerns, please acknowledge this and adjust your scores accordingly. If not, please let them know which concerns remain and if you have any follow-up questions. Your thoughtful feedback is crucial for helping the authors improve their work and advancing the field.

I realize this is a busy time and really appreciate your effort.

Best Regards
Area Chair

---

### Note · Authors · 2025-08-14

Dear Area Chair and Reviewers,

We sincerely appreciate your efforts in coordinating the reviews of our manuscript. Below, we summarize our work and key discussions from the rebuttal period.

Our work introduces a novel framework for distilling the text anonymization capabilities of large, proprietary models into smaller, locally run language models. The framework leverages adversarial interactions between an LLM anonymizer and an inference model to collect trajectories of anonymized texts, inferred private attributes, and utility evaluations. These trajectories are then used for supervised fine-tuning and direct preference optimization, distilling both anonymization and critique capabilities into a single model. This enables effective anonymization without relying on large, external models during inference. Experiments show that an 8B model can achieve competitive or superior privacy-utility trade-offs compared to GPT-4 anonymizers.

We appreciate the reviewers’ thoughtful comments and are encouraged by their overall positive assessment of our work. Reviewers recognized our work as a significant and original contribution (6CMa), highlighting its practical relevance in enabling private, efficient anonymization with a small, local model (wVoE, 6CMa). They also cited the strong empirical evaluation, with varied datasets, robust baselines, and ablation studies, as a key strength providing convincing evidence for our method’s effectiveness (wVoE, 6CMa, SaK9). Additionally, reviewers found our work innovative, well-written, and clearly explained (wVoE, 6CMa, hwtC).

We also value the constructive suggestions regarding further empirical validation with additional frontier models as judges, different model families, and more discussion of latency, privacy-utility trade-offs, and other relevant aspects of our work. In response, we have
- conducted additional evaluations with both human evaluators and frontier models, Claude Sonnet 4 and Gemini-2.5-Flash,
- assessed additional model families including Mistral and Phi models, and
- included detailed discussions on latency, privacy-utility trade-offs, and our contributions relative to prior work.

These additional results and discussions will be fully incorporated into the final version, which we believe will further strengthen the paper’s clarity and impact.

Thank you again for your careful consideration.

Warm regards,

Authors

---

### Decision · Program_Chairs · 2025-09-17

**Decision:**

Accept (poster)

**Comment:**

The authors propose a method for distilling the anonymization capabilities of LLMs into smaller models. The framework is based on adversarial interactions between an anonymizer and an inference model. Reviewers appreciated the motivation and methodology, and were generally positive about the work. They brought up several questions about utility degradation of the smaller models, as well as how well the approach generalizes to other domains. The authors responded to these questions in the rebuttal phase, clarifying several issues. The authors also conducted a human evaluation to compare human evaluations of textual quality relative to LLM-as-a-judge for evaluating utility. While the human study is appreciated, I would personally recommend the authors add more detail to their study description. It is currently unclear how the coders were assigned questions, whether multiple human subjects coded the same phrases, and how inter-rater disagreements were handled.

Overall, the work addresses a real and pressing problem, and it does so with some methodological innovations. Incorporating the suggestions of the reviewers, I expect this paper will be of interest to the NeurIPS audience.